# Simple and Nearly-Optimal Sampling for Rank-1 Tensor Completion via Gauss-Jordan

**Alejandro Gomez-Leos**                                          *alexgomezleos@utexas.edu*
*Department of Electrical and Computer Engineering*
*University of Texas at Austin*

**Oscar López**                                                              *lopezo@fau.edu*
*Harbor Branch Oceanographic Institute*
*Florida Atlantic University*

**Reviewed on OpenReview:** *https://openreview.net/forum?id=ggAphfUt1J*

## Abstract

We revisit the sample and computational complexity of the rank-1 tensor completion problem in $\otimes_{i=1}^N \mathbb{R}^d$, given a uniformly sampled subset of entries. We present a characterization of the problem which reduces to solving a pair of random linear systems. For example, when $N$ is a constant, we prove it requires no more than $m = O(d^2 \log d)$ samples and runtime $O(md^2)$. Moreover, we show that a broad class of algorithms require $\Omega(d \log d)$ samples, even under higher-rank scenarios. In contrast, existing upper bounds on the sample complexity are at least as large as $d^{1.5} \mu^{\Omega(1)} \log^{\Omega(1)} d$, where $\mu$ can be $\Theta(d)$ in the worst case. Prior work obtained these looser guarantees in higher-rank versions of our problem, and tend to involve more complicated algorithms.

## 1 Introduction

Tensor completion is a higher-order generalization of the well-known matrix completion problem (Candes & Tao, 2010; Candes & Recht, 2012). More precisely, an $N$-fold, order $d$ tensor $\mathbf{U}$ is a multi-dimensional array whose entries $\mathbf{U}_{(i_1,\ldots,i_N)}$ are specified by an ordered tuple of $N$ indices, each in $[d] := \{1,\ldots,d\}$. We denote $\otimes_{i=1}^N \mathbb{R}^d$ as the set of all such tensors. Stated loosely, the problem is to recover the entirety of a tensor $\mathbf{U}$ observing only a small subset of its entries.

If $\mathbf{U}$ is arbitrary, then this task is impossible without observing all $d^N$ entries of $\mathbf{U}$. As in matrix completion, the situation becomes interesting when $\mathbf{U}$ has a *low-rank* structure. Conceptually, this means $\mathbf{U}$'s entries are described by an interaction of a small number of variables—far fewer than $d^N$. In practice, the framework of tensor completion enjoys several applications in diverse areas such as recommender systems (Nguyen & Uhlmann, 2023), image and video processing (Wang et al., 2025), and bioinformatics (Liu et al., 2022). We refer to Kolda & Bader (2009); Song et al. (2019) for an excellent overview of other applications.

In this work, we revisit the problem of rank-1 tensor completion. A tensor $\mathbf{U} \in \otimes_{i=1}^N \mathbb{R}^d$ is said to be rank-1 if there exists $\{\mathbf{u}_1,\ldots,\mathbf{u}_N\} \subseteq \mathbb{R}^d$ such that

$$\mathbf{U}_{(i_1,i_2,\ldots,i_N)} = (\mathbf{u}_1)_{i_1} \cdot (\mathbf{u}_2)_{i_2} \cdot \ldots \cdot (\mathbf{u}_N)_{i_N} \qquad \forall (i_1, i_2 \ldots, i_N) \in [d]^N, \tag{1}$$

or $\mathbf{U} = \mathbf{u}_1 \otimes \cdots \otimes \mathbf{u}_N$, where $\otimes$ denotes the vector outer product. Perhaps the most fundamental variant, we consider the setting in which the estimation algorithm only has access to uniformly drawn entries and is required to be correct with bounded error probability. Specifically, we study the following:

> **Problem: Rank-1 Tensor Completion**
> **Input:** $m$ uniform samples (with replacement) from entries of rank-1 tensor $\mathbf{U} \in \otimes_{i=1}^{N} \mathbb{R}^d$
> **Output:** Oracle access to $\hat{\mathbf{U}}$ where $\hat{\mathbf{U}} = \mathbf{U}$ w.p. (with probability) $\geq 2/3$

Note that the choice of 2/3 is by convention rather than technicalities.[1] Given an algorithm for the above consuming $m$ samples, one can use the standard method of amplification to obtain an algorithm using $O(m \log(\frac{1}{\delta}))$ samples and succeeding with probability $\geq 1 - \delta$. Specifically, one would re-run the sampling procedure and algorithm on the same input $\mathbf{U}$ and, for each entry, take majority across the outputs.

We are interested in the sample and time complexity of the above as dependent on $d$ and $N$. Our contributions include a simple algorithm—applicable to rank-1 tensors with nonzero entries—and lower bounds against a broad class of estimation algorithms.

Our main results are summarized as follows, assuming infinite precision arithmetic throughout. Suppose $\mathbf{U}$ is an arbitrary rank-1 tensor with nonzero entries, and $N \asymp 1$ is a constant independent of $d$.[2]

- There exists an algorithm that solves **Rank-1 Tensor Completion** which draws $m \lesssim d^2 \log d$ samples and can be implemented in time $\lesssim md^2$.

- Moreover, $m \gtrsim d \log d$ samples are information-theoretically necessary for **Rank-1 Tensor Completion** to be solvable.

The full results are detailed in Theorem 1.3, Theorem 1.5, and Corollary 1.6. The main technical novelty of our work is a reduction between **Rank-1 Tensor Completion** and a particular matrix sketch problem, detailed in Section 2. In addition to revealing an equivalence between the sample complexities of each problem, the algorithm therein serves as a means of achieving a novel analysis of the tensor completion problem.

A key challenge in our approach is that the matrix sketch must be performed over the field $\mathbb{F}_2$—a problem which lacks a thorough study in the previous literature. Over $\mathbb{R}$, matrix sketching can be studied using tools from leverage score sampling (e.g. Cohen et al. (2015)). Unfortunately, these tools require the additive structure of $\mathbb{R}$, relying on matrix Chernoff bounds. We address this challenge by characterizing the sampling process as a random walk on a graph related to the matrix's rowspace. In this way, we relate the number of samples used by the algorithm to the walk's mixing time. This argument culminates in Lemma 2.2, which plays a key role in Theorem 1.3.

For higher-rank tensor completion problems, it is well-known that the complexities above are impacted by the *incoherence* $\mu$ of the input tensor class—perhaps surprisingly, our bounds have no such dependence. This observation seems to have gone undescribed in the current literature. In effect, this work serves to elucidate the sample complexity gap between the rank-1 and general rank problem variants. To properly contextualize these points, we review some basic definitions.

**Definition 1.1** (e.g. Cai et al. (2019); Singh et al. (2020)). *The rank of a tensor $\mathbf{U} \in \otimes_{i=1}^{N} \mathbb{R}^d$ is the minimum integer such that $\mathbf{U} = \sum_{k=1}^{r} \lambda_k \mathbf{u}_{1,k} \otimes \cdots \otimes \mathbf{u}_{N,k}$ holds for vectors $\{\mathbf{u}_{1,k}, \ldots, \mathbf{u}_{N,k}\}_{k=1}^{r} \subseteq \mathbb{R}^d$ and scalars $\{\lambda_k\}_{k=1}^{r} \subseteq \mathbb{R}$.*

**Definition 1.2** (e.g. Liu & Moitra (2020); Singh et al. (2020)). *A rank-r tensor is called $\mu$-incoherent if each of the following subspaces are $\mu$-incoherent*

$$\text{span}\left\{ \frac{\mathbf{u}_{1,k}}{\|\mathbf{u}_{1,k}\|_2}, \frac{\mathbf{u}_{2,k}}{\|\mathbf{u}_{2,k}\|_2}, \ldots, \frac{\mathbf{u}_{N,k}}{\|\mathbf{u}_{N,k}\|_2} \right\}, \quad k = 1, 2, \ldots, N. \tag{2}$$

In the rank-1 case, $\mathbf{U} = \mathbf{u}_1 \otimes \cdots \otimes \mathbf{u}_N$ being $\mu$-incoherent simply means $\|\mathbf{u}_i\|_{\infty} \leq \sqrt{\mu/d} \quad \forall i \in [d]$ (e.g. Liu & Moitra (2020); Singh et al. (2020)).

---

[1] Just as in, for example, the well-known definitions of complexity classes BPP and RP.

[2] $d \gg N$ in practice, with tensors of $N > 5$ rarely seen in applications. Thus, we focus on dependence in $d$.

**Prior Work**   The study of both rank-1 tensor completion and its sample complexity is a novel pairing. Therefore, in this section we trace the literature separately from each angle.

On one hand, our problem is a basic case of the well-studied exact tensor completion problem under uniform sampling. As such, we describe the prior bounds obtained for the rank-1 case. We note that some of these bounds are for the *expected* sample complexity under the Bernoulli model (Candes & Tao, 2010). Rest assured, the comparison is not that imprecise due to standard Chernoff bounds for binomial random variables.

Popularly, the earliest guarantees for matrices ($N = 2$) were given in Candes & Tao (2010); Recht (2011); Candes & Recht (2012). Notably, these exhibited a strong dependence of $\mu$ on the sample complexity, culminating in Chen (2015) establishing that $\lesssim d\mu \log^2 d$ samples suffice for $d^{-\Theta(1)}$ failure probability tolerance. For cubic tensors ($N = 3$), the works Jain & Oh (2014); Xia & Yuan (2019); Liu & Moitra (2020) indicate a scaling increase to $\lesssim d^{1.5}\mu^{O(1)} \log^{O(1)} d$ for a comparative failure tolerance. The theory for $N \geq 4$ has received much less attention, with the first results from Krishnamurthy & Singh (2013), although for adaptively chosen entries. These scaled as $\lesssim d\mu^{O(N)} N^{O(1)} \log \frac{d}{\delta}$ for tolerance $d\delta$. Under varying structural assumptions, the works Montanari & Sun (2018); Stephan & Zhu (2024); Haselby et al. (2024) make significant advancements, but still with dependence on $\mu^{O(1)}$. Since $\mu$ can be $\Theta(d)$ in the worst case, this substantiates our assertion that our above results elucidate a complexity gap between **Rank-1 Tensor Completion** and these more generic problems.[3] We emphasize that these works do not focus on rank-1 tensors, so it is reasonable to anticipate the improvements established in this work.

On the other hand, a distinct series of works Király et al. (2015); Kahle et al. (2017); Rendon Jaramillo (2018); Singh et al. (2020); Zhou et al. (2024) have focused on the *completability* of a fixed, partially observed rank-1 tensor (i.e. the solution set for the polynomial system in equation 1). Perhaps as a testament to the non-triviality of this problem, these works invoke advanced tools from algebraic geometry and matroid theory (Király et al., 2015; Rendon Jaramillo, 2018). We emphasize these do not study the sampling aspect of our problem.

The algorithms of these prior works tend to be quite intricate, albeit specialized for the harder problem of general rank tensor completion. A few popular methods are alternating minimization (Jain & Oh, 2014; Xia & Yuan, 2019; Liu & Moitra, 2020) and semidefinite programming / sum-of-squares hierarchies (Chen et al., 2015; Potechin & Steurer, 2017; Zhou et al., 2024), amongst others. For a more complete overview, we refer to Cai et al. (2019); Liu & Moitra (2020); Haselby et al. (2024).

By contrast, in this work we describe a linear algebraic characterization, which was independently observed by Singh et al. (2020) and Stephan & Zhu (2024). However, the former did not leverage this to study the statistical hardness of our problem, and the latter's discussion was limited to symmetric Boolean tensors. In this work we address both of these limitations, showing **Rank-1 Tensor Completion** admits an extremely simple algorithm, whose main bottleneck is solving a $\tilde{O}(d^2) \times \tilde{O}(d)$ system over $\mathbb{F}_2$. As a consequence, the toolbox for our analysis only consists of elementary linear algebraic and probabilistic arguments. Before detailing our results, we overview the notation to be used throughout this work.

**Notation**   Throughout, $\mathbb{F}_2$ denotes the finite field of order 2. For $\alpha, \beta \in \mathbb{R}$, we write $\alpha \lesssim \beta$ if $\alpha \leq C\beta$ for some absolute constant $C > 0$, and $\alpha \asymp \beta$ if $\alpha \lesssim \beta$ and $\beta \lesssim \alpha$. We reserve bold \mathsfit for tensors (e.g. **U**), upper-case \mathbf for matrices (e.g. $\mathbf{U}$), and lower-case \mathbf for vectors (e.g. $\mathbf{u}$). All vectors are conventionally column vectors, unless otherwise stated. We denote $\mathbf{e}_k$ for $k \in [d]$ as the indicator row vector for $[d]$, interpreting the $\mathbf{e}_k$'s having entries in $\mathbb{F}_2$ or $\mathbb{R}$, wherever clear from context. For two row vectors $\mathbf{x}$ and $\mathbf{y}$, we denote $[\mathbf{x}, \mathbf{y}]$ as their concatenation. For a $n_1 \times n_2$ matrix $\mathbf{A}$ and the vector $\mathbf{b}$ with $n_1$ elements, we denote their induced linear system over a field $\mathbb{K}$ by the augmented matrix $[\mathbf{A} \mid \mathbf{b}]_{\mathbb{K}}$, regardless of whether the system is consistent or not. For a scalar-valued function $f$ applied to a tensor, matrix, or vector, we mean entry-wise. For a matrix $\mathbf{B}$, we denote $\mathbf{B}_i$ as its $i^{th}$ row. We denote the rowspace of a matrix $\mathbf{B}$ over field $\mathbb{K}$ by $\text{row}_{\mathbb{K}}(\mathbf{B})$. The Frobenius norm of a tensor is given by $\|\mathbf{U}\|_F^2 = \sum_{(i_1,\dots,i_N)} \mathbf{U}_{(i_1,\dots,i_N)}^2$. We denote the set of all $N$-fold, order $d$ tensors with nonzero entries as $\otimes_{i=1}^N \mathbb{R}_{\neq 0}^d$. For random variables $X$ and

---

[3]Two lower bounds are given in Candes & Tao (2010); Singh et al. (2020) (Theorem 1.7 and Theorem 5, resp.). These seem to suggest a multiplicative $\mu^{\Omega(N)}$ is missing from our upper bound, but their hard instances do not apply.

$Y$ admitting a joint distribution, $H(Y \mid X)$ denotes the conditional entropy of random variable $Y$ given $X$. For two scalar-valued functions $f$ and $g$, $f \circ g$ denotes their composition.

**Our Results**   In this work, we show the following.

**Theorem 1.3.** *If $\mathbf{U} \in \otimes_{i=1}^{N} \mathbb{R}_{\neq 0}^{d}$ is a rank-1 tensor, then there exists an algorithm solving **Rank-1 Tensor Completion** using $m \lesssim (dN)^2 \log d$ uniformly drawn entries, and can be implemented in time $\lesssim qN + md^2$, where $q$ is the number of queried entries of $\mathbf{U}$.*

To prove it, in Section 2 we show every rank-1 tensor in $\otimes_{i=1}^{N} \mathbb{R}_{\neq 0}^{d}$ is in correspondence with a pair of linear systems—one over $\mathbb{F}_2$, and the other over $\mathbb{R}$. The former system "encodes information" about the signs of the tensor's entries, whereas the latter system does so for their magnitude. In a precise sense, we show that the "information" revealed by sampling an entry corresponds to the "information" encoded by one row of the associated augmented matrix. These systems are overdetermined, so it suffices to obtain just a small subset of their rows. In essence, this characterization illuminates that the sample complexity can be studied through a matrix row sketch problem over $\mathbb{F}_2$. Unfortunately, the structure of these linear systems also imply that the resulting algorithm cannot be trivially extended to higher-rank versions of our problem. Nonetheless, this reduction is highly valuable in our problem setting.

We also present a lower bound against a broad class of estimation algorithms, defined below, which matches the above result up to a factor in $d$ when $N \asymp 1$. As a result, the proposed analysis only shows the above is nearly-optimal in $d$, although the analysis is indeed optimal if $\mathbf{U}$ is required to be unsigned—we elaborate on this subtle point at the end of Section 2. For the same reason, we believe this is an artifact of our particular upper bound technique, rather than the high-level approach. We leave its resolution as an open problem.

**Definition 1.4** (randomized estimation algorithm)**.** *We say $\mathcal{A}$ is a randomized estimation algorithm for a random tensor $\mathbf{U}$, random observation subset $S \subseteq [d]^N$, and internal random bits $B \in \{0,1\}^{\infty}$ if its output is determined by the outcome of all three, i.e. $\mathcal{A}(\mathbf{U}, S, B)$ satisfies $H(\mathcal{A}(\mathbf{U}, S, B) \mid \mathbf{U}, S, B) = 0$. Moreover, we say the estimator draws $m$ samples if $|S| = m$.*

**Theorem 1.5.** *Let $\sigma > 0$. There exists a distribution $\mathcal{D}$ over rank-1 tensors in $\{\mathbf{U} \in \otimes_{i=1}^{N} \mathbb{R}_{\neq 0}^{d} : \|\mathbf{U}\|_{\infty} \leq \sigma\}$ and absolute constants $n_0 \in \mathbb{N}$ and $C > 0$ such that, for $dN \geq n_0$, any randomized estimator $\hat{\mathbf{U}}$ drawing less than $Cd \log dN$ samples from $\mathbf{U} \sim \mathcal{D}$ suffers error $\|\hat{\mathbf{U}} - \mathbf{U}\|_F \geq \sigma \sqrt{d^{N-1}}$ w.p. $> 1/3$. Therefore, there is no algorithm to solve **Rank-1 Tensor Completion** when $m$ is below this threshold.*

Lower bounds in the field are usually stated in a different sense—that if too few samples are taken, then there would likely exist distinct tensors agreeing on the observed entries. The standard conclusion is that any algorithm that solely bases its decision on the observed entries must fail (Candes & Tao, 2010; Krishnamurthy & Singh, 2013). However, this conclusion might be unsatisfying since it does not quantify the error, nor clarifies whether extra randomness is useful. This motivates the style of our bound.

Notably, Theorem 1.5 extends to higher-rank tensors, under a benign "consistency" assumption. Due to its highly technical statement, we expound Assumption 6.2 in Section 6. To summarize, it roughly asserts that the estimator's error is weakly increasing for perturbations to its input. In some sense, this class captures algorithms whose precision worsens with higher "noise". For such algorithms, the following lower bound holds.

**Corollary 1.6.** *Let $\sigma > 0$. Let $n_0$ and $C$ be as in Theorem 1.5 and assume $dN \geq n_0$. If a Hadamard matrix of order $d$ exists[4], then there exists a distribution $\mathcal{D}$ over tensors of rank $\gtrsim r$ in $\{\mathbf{U} \in \otimes_{i=1}^{N} \mathbb{R}_{\neq 0}^{d} : \|\mathbf{U}\|_{\infty} \leq \sigma\}$ such that any randomized estimator $\hat{\mathbf{U}}$ satisfying Assumption 6.2 and drawing less than $Cd \log dN$ samples suffers error $\|\hat{\mathbf{U}} - \mathbf{U}\|_F \gtrsim \sigma \sqrt{d^{N-1}}$ w.p. $> 1/3$.*

Concluding our main results, we delve into a characterization of rank-1 tensors in $\otimes_{i=1}^{N} \mathbb{R}_{\neq 0}^{d}$ which forms the foundation of our algorithm and its analysis.

---

[4]Recall such a matrix is $d \times d$, has entries in $\{\pm 1\}$, and admits mutually orthogonal columns.

## 2 Nonzero Rank-1 Tensors in Exponential Form

In this section, we describe a characterization of rank-1 tensors in $\otimes_{i=1}^N \mathbb{R}_{\neq 0}^d$, which is at the core of Algorithm 1. Suppose that $\mathbf{U} \in \otimes_{i=1}^N \mathbb{R}_{\neq 0}^d$ satisfies equation 1 for some column vectors $\mathbf{u}_1, \ldots, \mathbf{u}_N \subseteq \mathbb{R}^d$. Let us define $\mathbf{x} \in \mathbb{R}^{dN}$ as

$$\mathbf{x} := \begin{pmatrix} \mathbf{u}_1 \\ \mathbf{u}_2 \\ \vdots \\ \mathbf{u}_N \end{pmatrix} = \begin{pmatrix} \begin{pmatrix} (\mathbf{u}_1)_1 \\ \vdots \\ (\mathbf{u}_1)_d \end{pmatrix} \\ \vdots \\ \begin{pmatrix} (\mathbf{u}_N)_1 \\ \vdots \\ (\mathbf{u}_N)_d \end{pmatrix} \end{pmatrix}. \tag{3}$$

Setting this aside, we focus on $\mathbf{U}$, which we can re-write as follows for all $(i_1, \ldots, i_N) \in [d]^N$.

$$\mathbf{U}_{(i_1, i_2, \ldots, i_N)} = \text{sign}\left(\prod_{\ell=1}^N (\mathbf{u}_\ell)_{i_\ell}\right) \left|\prod_{\ell=1}^N (\mathbf{u}_\ell)_{i_\ell}\right|$$

$$= \left(\prod_{\ell=1}^N \text{sign}\left((\mathbf{u}_\ell)_{i_\ell}\right)\right) \left(\exp\left(\sum_{\ell=1}^N \log |(\mathbf{u}_\ell)_{i_\ell}|\right)\right)$$

$$:= \mathbf{U}'_{(i_1, i_2, \ldots, i_N)} \exp\left(\mathbf{U}''_{(i_1, i_2, \ldots, i_N)}\right) \tag{4}$$

Noting that $\mathbf{U}'_{(i_1, i_2, \ldots, i_N)} = -1$ iff an odd number of the variables $(\mathbf{u}_1)_{i_1} \ldots (\mathbf{u}_N)_{i_N}$ are negative, we observe that the entry $\mathbf{U}'_{(i_1, i_2, \ldots, i_N)}$ resembles the parity function on the sign of these variables. Hence, using an appropriate 1-1 transformation $\varphi$ between $\{\pm 1\}$ and $\{0,1\}$ (e.g. $\varphi(z) := -\frac{1}{2}(z-1)$), we obtain that the $\{\mathbf{u}_\ell\}_{\ell=1}^N$ solve the following linear systems over $\mathbb{F}_2$ and $\mathbb{R}$:

$$\sum_{\ell=1}^N (\varphi \circ \text{sign})((\mathbf{u}_\ell)_{i_\ell}) = (\varphi \circ \text{sign})\left(\mathbf{U}_{(i_1, i_2, \ldots, i_N)}\right) \mod 2$$

$$\sum_{\ell=1}^N (\log \circ \text{abs})((\mathbf{u}_\ell)_{i_\ell}) = (\log \circ \text{abs})\left(\mathbf{U}_{(i_1, i_2, \ldots, i_N)}\right).$$

We now describe the coefficient matrix of these systems. Let $\pi : [d]^N \to [d^N]$ denote a fixed bijection throughout. Defining each row as $\mathbf{A}_{\pi(i_1, \ldots, i_N)} := [\mathbf{e}_{i_1}, \mathbf{e}_{i_2}, \ldots, \mathbf{e}_{i_N}]$ for all $(i_1, \ldots, i_N) \in [d]^N$, we have

$$\mathbf{A}_{\pi(i_1, \ldots, i_N)}(\varphi \circ \text{sign})(\mathbf{x}) = (\varphi \circ \text{sign})\left(\mathbf{U}_{(i_1, i_2, \ldots, i_N)}\right) \mod 2$$

$$\mathbf{A}_{\pi(i_1, \ldots, i_N)}(\log \circ \text{abs})(\mathbf{x}) = (\log \circ \text{abs})\left(\mathbf{U}_{(i_1, i_2, \ldots, i_N)}\right).$$

Hence, there is a unique 1-1 tensor-to-vector map $\text{vec}_\pi$ such that the above is equivalent to

$$\mathbf{A}(\varphi \circ \text{sign})(\mathbf{x}) = (\varphi \circ \text{sign})(\text{vec}_\pi \mathbf{U}) \mod 2$$

$$\mathbf{A}(\log \circ \text{abs})(\mathbf{x}) = (\log \circ \text{abs})(\text{vec}_\pi \mathbf{U}).$$

To avoid overloading notation throughout, we let $f := \varphi \circ \text{sign} \circ \text{vec}_\pi$ and $\tilde{f} := \log \circ \text{abs} \circ \text{vec}_\pi$, so

$$\mathbf{A}(\varphi \circ \text{sign})(\mathbf{x}) = f(\mathbf{U}) \mod 2$$

$$\mathbf{A}(\log \circ \text{abs})(\mathbf{x}) = \tilde{f}(\mathbf{U}).$$

Notably, $\mathbf{A}$'s rows enumerate all $d^N$ possible vectors of size $dN$ obtained by concatenating $N$ row vectors from $\{\mathbf{e}_k\}_{k \in [d]}$ (each row corresponds to a unique tensor entry). This observation enables a simple proof of a result we use extensively—that $\mathbf{A}$ has the same rank considered as matrix over $\mathbb{F}_2$ or $\mathbb{R}$.

**Lemma 2.1.** *The matrix $\mathbf{A}$ satisfies* $\text{rank}_{\mathbb{F}_2}(\mathbf{A}) = \text{rank}_{\mathbb{R}}(\mathbf{A}) = dN - (N-1)$.

*Proof.* See Section 3. □

In summary of the previous observations, we have the following.

**Observation 1.** *For any tensor* $\mathbf{U} \in \otimes_{i=1}^{N} \mathbb{R}_{\neq 0}^{d}$,

$$\mathbf{U} \text{ is rank-1} \implies \text{linear systems } [\, \mathbf{A} \mid f(\mathbf{U}) \,]_{\mathbb{F}_2} \text{ and } [\, \mathbf{A} \mid \tilde{f}(\mathbf{U}) \,]_{\mathbb{R}} \text{ are both consistent.} \tag{5}$$

*Furthermore, if for a rank-1 tensor* $\mathbf{T} \in \otimes_{i=1}^{N} \mathbb{R}_{\neq 0}^{d}$ *the joint solution sets of*

$$\left( [\, \mathbf{A} \mid f(\mathbf{T}) \,]_{\mathbb{F}_2}, [\, \mathbf{A} \mid \tilde{f}(\mathbf{T}) \,]_{\mathbb{R}} \right) \text{ and } \left( [\, \mathbf{A} \mid f(\mathbf{U}) \,]_{\mathbb{F}_2}, [\, \mathbf{A} \mid \tilde{f}(\mathbf{U}) \,]_{\mathbb{R}} \right) \tag{6}$$

*are equivalent, then* $\mathbf{U} = \mathbf{T}$.

Indeed, equation 6 is the consequence of the following procedure. Supposing we had access to a solution of $\mathbf{y}_1$ of $[\, \mathbf{A} \mid f(\mathbf{U}) \,]_{\mathbb{F}_2}$, and a solution $\mathbf{y}_2$ of $[\, \mathbf{A} \mid \tilde{f}(\mathbf{U}) \,]_{\mathbb{R}}$, then we could recover the tensor from

$$\mathbf{U} = \mathrm{vec}_{\pi}^{-1}(\varphi^{-1}(\mathbf{A}\mathbf{y}_1) \odot \exp(\mathbf{A}\mathbf{y}_2))$$

where $\odot$ denotes the Hadamard product. Therefore, the crux of the issue is accessing these solutions. Fortunately, as we'll formalize, this can be achieved by observing sufficiently many random entries of $\mathbf{U}$.

To preface, for each subset $S \subseteq [d]^N$, we define the *row selection matrix* $\mathbf{D}_S$ as the matrix such that

$$\mathbf{D}_S \mathbf{A} = \begin{pmatrix} \vdots \\ \mathbf{A}_{\pi(i_1,\ldots,i_N)} \\ \vdots \end{pmatrix} \quad \forall (i_1, \ldots, i_N) \in S, \tag{7}$$

where the rows adhere to the ordering induced by $\pi$. We similarly define $\mathbf{D}_{\bar{S}}$ for the complement $\bar{S} := [d]^N \setminus S$. As a result, there is always a row permutation matrix $\mathbf{P}_S \in \{0,1\}^{d^N \times d^N}$ where

$$\mathbf{P}_S \mathbf{A} = \begin{pmatrix} \mathbf{D}_S \mathbf{A} \\ \mathbf{D}_{\bar{S}} \mathbf{A} \end{pmatrix}. \tag{8}$$

As we've established, since each entry corresponds with a particular row of augmented systems, we have the next observation.

**Observation 2.** *Let* $\mathbf{U} \in \otimes_{i=1}^{N} \mathbb{R}_{\neq 0}^{d}$ *be rank-1 tensor, and let* $S \subseteq [d]^N$. *Then the subset of entries* $\{\mathbf{U}_{(i_1,\ldots,i_N)}\}_{(i_1,\ldots,i_N) \in S}$ *are in correspondence with the pair of linear systems*

$$[\, \mathbf{D}_S \mathbf{A} \mid \mathbf{D}_S f(\mathbf{U}) \,]_{\mathbb{F}_2}, \quad [\, \mathbf{D}_S \mathbf{A} \mid \mathbf{D}_S \tilde{f}(\mathbf{U}) \,]_{\mathbb{R}}. \tag{9}$$

Combining Observation 2 and Observation 1, it is immediate that $S$ just needs to satisfy that equation 9 has the same joint solution set as $\left( [\, \mathbf{A} \mid f(\mathbf{U}) \,]_{\mathbb{F}_2}, \quad [\, \mathbf{A} \mid \tilde{f}(\mathbf{U}) \,]_{\mathbb{R}} \right)$ to complete the tensor via the aforementioned procedure.

In Section 5, we show the sufficient conditions are $\mathrm{row}_{\mathbb{F}_2}(\mathbf{D}_S \mathbf{A}) = \mathrm{row}_{\mathbb{F}_2}(\mathbf{A})$ and $\mathrm{row}_{\mathbb{R}}(\mathbf{D}_S \mathbf{A}) = \mathrm{row}_{\mathbb{R}}(\mathbf{A})$. In other words, given that these hold, any joint solution to equation 9 yields a joint solution of the overall systems. The exact procedure is described in the pseudocode for Algorithm 1.

Hence, we can use Algorithm 1 to solve **Rank-1 Tensor Completion** by simply running it for a randomly drawn $S$. Clearly then, we only need to show these conditions hold with sufficient probability. To this end, we prove the following lemma, which states $O(d^2 \log d)$ samples suffice, although we conjecture it can be improved to $O(d \log d)$. Since the result below is crucial for Theorem 1.3, we elaborate on this conjecture following our overview of the proof technique.

**Lemma 2.2.** *Let* $S$ *be the subset of indices induced by* $m$ *uniformly drawn rows of* $\mathbf{A}$, *with replacement. Then,* $m \lesssim (dN)^2 \log d$ *samples suffice to ensure both simultaneously hold w.p.* $\geq 2/3$.

$$\mathrm{row}_{\mathbb{F}_2}(\mathbf{D}_S \mathbf{A}) = \mathrm{row}_{\mathbb{F}_2}(\mathbf{A}), \quad \mathrm{row}_{\mathbb{R}}(\mathbf{D}_S \mathbf{A}) = \mathrm{row}_{\mathbb{R}}(\mathbf{A}) \tag{10}$$

---

**Algorithm 1** Rank-1 Linear System Solver

---

**Input:** Tensor entries $\{\mathbf{U}_{(i_1,\ldots,i_N)}\}_{(i_1,\ldots,i_N)\in S}$ for $S \subseteq [d]^N$

**Output:** $\hat{\mathbf{U}}_{(i_1,\ldots,i_N)}$ for each desired $(i_1,\ldots,i_N) \in [d]^N$

  1: $\mathbf{y}_1 \leftarrow$ any solution $\mathbf{y}$ to $\mathbf{D}_S\mathbf{A}\mathbf{y} = \mathbf{D}_S f(\mathbf{U})$ over $\mathbb{F}_2$

  2: $\mathbf{y}_2 \leftarrow$ any solution $\mathbf{y}$ to $\mathbf{D}_S\mathbf{A}\mathbf{y} = \mathbf{D}_S \tilde{f}(\mathbf{U})$ over $\mathbb{R}$

  3: **return** $\hat{\mathbf{U}}_{(i_1,\ldots,i_N)} = \varphi^{-1}(\langle \mathbf{A}_{\pi(i_1,\ldots,i_N)}, \mathbf{y}_1\rangle) \exp(\langle \mathbf{A}_{\pi(i_1,\ldots,i_N)}, \mathbf{y}_2\rangle)$ for $(i_1,\ldots,i_N) \in [d]^N$

---

*Proof.* See Section 4. $\qquad\qquad\qquad\qquad\qquad\qquad\qquad\qquad\qquad\qquad\qquad\qquad\qquad\qquad\qquad\quad$ $\square$

At a high-level, our proof views $\mathrm{row}_{\mathbb{F}_2}(\mathbf{D}_S\mathbf{A})$ as the result of a sequentially constructed subspace $W$ of $\mathbf{A}$'s rowspace. We correspond the sample paths of this random process to trajectories on a Markov chain whose states are indexed by $(\dim W, W)$. Due to the first state coordinate, the chain jumps to a new state no more than $\mathrm{rank}_{\mathbb{F}_2}(\mathbf{A})$ times before hitting the absorbing state $(\mathrm{rank}_{\mathbb{F}_2}(\mathbf{A}), \mathrm{row}_{\mathbb{F}_2}(\mathbf{A}))$. As a result, the measure of the "bad" event $\{\mathrm{row}_{\mathbb{F}_2}(\mathbf{D}_S\mathbf{A}) \neq \mathrm{row}_{\mathbb{F}_2}(\mathbf{A})\}$ is given by the cumulative measure of the "bad" trajectories, i.e. those that stagnate and never hit the absorbing state.

It turns out to be simple to prove the chain self-loops w.p. $\leq 1 - 1/d$. Moreover, by pigeonholing, every "bad" trajectory of length $T > \mathrm{rank}_{\mathbb{F}_2}(\mathbf{A})$ has $\Omega(T - \mathrm{rank}_{\mathbb{F}_2}(\mathbf{A}))$ self-loops. Thus, the measure of each "bad" trajectory shrinks exponentially with rate rate $\frac{1}{d}\Omega(T - \mathrm{rank}_{\mathbb{F}_2}(\mathbf{A}))$. A counting argument shows there are no more than $d^{N \, \mathrm{rank}_{\mathbb{F}_2}(\mathbf{A})}$ "bad" trajectories, from which we show $O(d \log d^{N \, \mathrm{rank}_{\mathbb{F}_2}(\mathbf{A})}) = O((dN)^2 \log d)$ samples suffice. To handle the other condition (over $\mathbb{R}$), we show it readily follows from the next well-known fact in tandem with Lemma 2.1.

**Fact 2.3.** *For any binary matrix* $\mathbf{B}$, $\mathrm{rank}_{\mathbb{F}_2}(\mathbf{B}) \leq \mathrm{rank}_{\mathbb{R}}(\mathbf{B})$.

Therefore, it is clear that the bottleneck for improvement is the sample complexity of sketching $\mathbf{A}$ over $\mathbb{F}_2$: supposing that $\mathbf{U}$ was unsigned, then $\mathbf{A}$ would only need to be sketched as a real matrix. By well-known results in leverage score sampling (e.g. Cohen et al. (2015)), one only needs to observe $O(d \log d)$ rows of a rank $O(d)$ real matrix to recover its rowspace, if it has uniform leverage scores. And one can easily verify that $\mathbf{A}$ satisfies this condition. This implies that the algorithm would be optimal in $d$ for this special case, leaving no particular reason to believe the $\mathbb{F}_2$ sketch *requires* $\Omega(d^2 \log d)$. For the interested reader, we refer to Lemma 4.1 as the candidate for improvement.

In the remaining sections, we build up towards Theorem 1.3, beginning with establishment of $\mathbf{A}$'s rank in the next section.

## 3 Proof of Lemma 2.1

To give a preview, our strategy is to construct a considerably simpler matrix whose rowspace is identical to $\mathbf{A}$'s. We then show that this matrix has the claimed rank. Additionally, the structure of these matrices will be a useful reference to streamline the proofs in the next sections. We now provide details of the proof.

To start, fix an arbitrary row $[\mathbf{e}_{i_1},\ldots,\mathbf{e}_{i_N}] \in \{\mathbf{A}_1,\ldots,\mathbf{A}_{d^N}\}$, and let $\Phi \in \{0,1\}^{(d-1)N+1 \times dN}$ be the following $\mathbb{F}_2$-valued matrix:

$$\Phi := \left(\begin{array}{ccccc} \mathbf{e}_{i_1} & \mathbf{e}_{i_2} & \mathbf{e}_{i_3} & \cdots & \mathbf{e}_{i_N} \\ \hline \mathbf{e}_1 + \mathbf{e}_2 & \mathbf{0}_d & \mathbf{0}_d & \cdots & \mathbf{0}_d \\ \mathbf{e}_1 + \mathbf{e}_3 & \mathbf{0}_d & \mathbf{0}_d & \cdots & \mathbf{0}_d \\ \vdots & \vdots & \vdots & \ddots & \vdots \\ \mathbf{e}_1 + \mathbf{e}_d & \mathbf{0}_d & \mathbf{0}_d & \cdots & \mathbf{0}_d \\ \hline \mathbf{0}_d & \mathbf{e}_1 + \mathbf{e}_2 & \mathbf{0}_d & \cdots & \mathbf{0}_d \\ \mathbf{0}_d & \mathbf{e}_1 + \mathbf{e}_3 & \mathbf{0}_d & \cdots & \mathbf{0}_d \\ \vdots & \vdots & \vdots & \ddots & \vdots \\ \mathbf{0}_d & \mathbf{e}_1 + \mathbf{e}_d & \mathbf{0}_d & \cdots & \mathbf{0}_d \\ \hline \mathbf{0}_d & \mathbf{0}_d & \mathbf{e}_1 + \mathbf{e}_2 & \cdots & \mathbf{0}_d \\ \mathbf{0}_d & \mathbf{0}_d & \mathbf{e}_1 + \mathbf{e}_3 & \cdots & \mathbf{0}_d \\ \vdots & \vdots & \vdots & \ddots & \vdots \\ \mathbf{0}_d & \mathbf{0}_d & \mathbf{e}_1 + \mathbf{e}_d & \cdots & \mathbf{0}_d \\ \hline \vdots & \vdots & \vdots & \ddots & \vdots \\ \hline \mathbf{0}_d & \mathbf{0}_d & \mathbf{0}_d & \cdots & \mathbf{e}_1 + \mathbf{e}_2 \\ \mathbf{0}_d & \mathbf{0}_d & \mathbf{0}_d & \cdots & \mathbf{e}_1 + \mathbf{e}_3 \\ \vdots & \vdots & \vdots & \ddots & \vdots \\ \mathbf{0}_d & \mathbf{0}_d & \mathbf{0}_d & \cdots & \mathbf{e}_1 + \mathbf{e}_d \end{array}\right) := \begin{pmatrix} [\mathbf{e}_{i_1}, \ldots, \mathbf{e}_{i_N}] \\ \Phi_1 \\ \Phi_2 \\ \vdots \\ \Phi_N \end{pmatrix} \tag{11}$$

For the next claims, recall that $\mathbf{A}$'s rows consist of all $d^N$ possible vectors of size $dN$ obtained by concatenating $N$ row vectors from $\{\mathbf{e}_k\}_{k \in [d]}$.

**Lemma 3.1.** $\mathrm{row}_{\mathbb{F}_2}(\Phi) = \mathrm{row}_{\mathbb{F}_2}(\mathbf{A})$.

*Proof of Lemma 3.1.* By the above assertion, the first row of $\Phi$ is a row of $\mathbf{A}$. Consider any other row of $\Phi$. This row can be expressed as the sum $[\mathbf{e}_1, \mathbf{e}_1, \ldots, \mathbf{e}_1] + [\mathbf{e}_1, \ldots, \mathbf{e}_1, \mathbf{e}_k, \mathbf{e}_1 \ldots, \mathbf{e}_1] \mod 2$ for some $k \in [d]$—both of which are also rows of $\mathbf{A}$. Hence, each individual row is contained in $\mathrm{row}_{\mathbb{F}_2}(\mathbf{A})$, so $\mathrm{row}_{\mathbb{F}_2}(\Phi) \subseteq \mathrm{row}_{\mathbb{F}_2}(\mathbf{A})$. On the other hand, for an arbitrary row $[\mathbf{e}_{j_1}, \ldots, \mathbf{e}_{j_N}]$ of $\mathbf{A}$, we can write

$$[\mathbf{e}_{j_1}, \ldots, \mathbf{e}_{j_N}] = [\mathbf{e}_{i_1}, \ldots, \mathbf{e}_{i_N}] + \sum_{k=1}^{N} [\mathbf{0}_d, \ldots, \mathbf{0}_d, (\mathbf{e}_1 + \mathbf{e}_{i_k}) + (\mathbf{e}_1 + \mathbf{e}_{j_k}), \mathbf{0}_d, \ldots, \mathbf{0}_d] \mod 2.$$

Each of the summands are evidently in the rowspace of $\Phi$. Hence, $\mathrm{row}_{\mathbb{F}_2}(\mathbf{A}) \subseteq \mathrm{row}_{\mathbb{F}_2}(\Phi)$. $\square$

**Lemma 3.2.** $\dim \mathrm{row}_{\mathbb{F}_2}(\Phi) = dN - (N - 1)$

*Proof of Lemma 3.2.* We first establish that each submatrix $\Phi_i$ for $i \in [N]$ has full row rank. Without loss of generality, consider the rows of $\Phi_1$. Assume by contradiction $\mathbf{c} := (c_1, \ldots, c_{d-1}) \neq \mathbf{0}_{d-1} \in \mathbb{F}_2^{d-1}$ describes a trivial linear combination of them, i.e.

$$\left[\mathbf{e}_1 \sum_{i=1}^{d-1} c_i + \sum_{i=1}^{d-1} c_i \mathbf{e}_{i+1}, \mathbf{0}_d, \ldots, \mathbf{0}_d\right] = \mathbf{0}_{dN} \mod 2.$$

Since no subset of $\mathbf{e}_2, \ldots, \mathbf{e}_d$ sums to $\mathbf{0}_d$, $\sum_{i=1}^{d-1} c_i \mathbf{e}_{i+1}$ must be identically $\mathbf{0}_d$, but this necessitates that $\mathbf{c} = \mathbf{0}_{d-1}$, contradiction. An analogous argument applies for every submatrix. Thus, $\mathrm{rank}_{\mathbb{F}_2}(\Phi_1) = \cdots = \mathrm{rank}_{\mathbb{F}_2}(\Phi_N) = d - 1$.

It is easy to see that any linear combination of rows in $\Phi_1, \ldots, \Phi_N$ result in a nonzero vector. Hence, the stacked matrix $(\Phi_1; \Phi_2; \ldots; \Phi_N)$ constitutes a linearly independent set of size $(d-1)N$. We now elucidate

that the first vector is also linearly independent of this stacked matrix. To this end, assume by contradiction that there exists some linear combination of rows in $\Phi_1, \ldots, \Phi_N$ that sum to $[\mathbf{e}_{i_1}, \ldots, \mathbf{e}_{i_N}]$, with coefficients

$$\mathbf{c} := [\mathbf{c}^1, \mathbf{c}^2, \ldots, \mathbf{c}^N] = \left((c_1^1, \ldots, c_{d-1}^1), \ldots, ((c_1^N, \ldots, c_{d-1}^N)\right) \in \mathbb{F}_2^{(d-1)N}$$

Picking any $k \in [N]$, the $k^{th}$ *column-block* satisfies

$$\mathbf{e}_1 \sum_{i=1}^{d-1} c_i^k + \sum_{i=1}^{d-1} c_i^k \mathbf{e}_{i+1} = \mathbf{e}_{i_k} \mod 2.$$

If $\mathbf{e}_{i_k} = \mathbf{e}_1$, then the first sum dictates $c_1^k, \ldots, c_{d-1}^k$ must have odd parity and contributes a single bit overall. But then the second sum contributes an odd number of bits. This yields a mismatch between the parity on both sides. If $\mathbf{e}_{i_k} \neq \mathbf{e}_1$, then second sum must have all $c_i^k$'s as zero except for one of them, but then the first sum contributes a single bit, a contradiction for the same reason as before. Thus, the $(d-1)N + 1$ rows of $\Phi$ span a linear space of dimension $dN - (N-1)$.

$\square$

Lemma 3.2 and Lemma 3.1 together imply $\text{row}_{\mathbb{F}_2}(\mathbf{A}) = dN - (N-1)$. For easy reference, we state a trivial corollary following from the fact that $[\mathbf{e}_{i_1}, \ldots, \mathbf{e}_{i_N}]$ was initially picked arbitrarily.

**Corollary 3.3.** *For any $[\mathbf{e}_{i_1}, \ldots, \mathbf{e}_{i_N}] \in \{\mathbf{A}_1, \ldots, \mathbf{A}_{d^N}\}$ chosen to construct $\Phi$, the rows of $\Phi$ in equation 11 consist of a basis of $\text{row}_{\mathbb{F}_2}(\mathbf{A})$.*

To handle the case of $\mathbb{R}$, one can apply an almost identical proof to establish the analogous claims for a $\mathbb{R}$-valued matrix $\tilde{\Phi}$, which instead contains rows of the form $[\mathbf{0}_d, \ldots, \mathbf{0}_d, \mathbf{e}_1 - \mathbf{e}_{i_k}, \mathbf{0}_d, \ldots, \mathbf{0}_d]$. We relegate the full matrix description to the Appendix B. Following this, the proof of Lemma 2.1 is complete.

## 4 Proof of Lemma 2.2

To prove Lemma 2.2, we can view the rowspace of $\mathbf{D}_S \mathbf{A}$ as the the cumulative span of the random variable sequence $Y_1, Y_2, \ldots, Y_m$ where $Y_t \overset{iid}{\sim} \text{Unif}(\{\mathbf{A}_1, \ldots, \mathbf{A}_{d^N}\})$.

Importantly, by Fact 2.3 and Lemma 2.1, whenever we have $\text{row}_{\mathbb{F}_2}(\mathbf{D}_S \mathbf{A}) = \text{row}_{\mathbb{F}_2}(\mathbf{A})$ we also have $\text{row}_{\mathbb{R}}(\mathbf{D}_S \mathbf{A}) = \text{row}_{\mathbb{R}}(\mathbf{A})$. Therefore, to prove our choice of $m$ suffices it is enough to show $\dim \text{span}\{Y_1 \ldots Y_m\} = \text{rank}_{\mathbb{F}_2}(\mathbf{A})$ w.p. $\geq 2/3$. Before proceeding to the proof, we assume the next claim holds, which we verify in the sequel.

**Lemma 4.1.** *Suppose $W$ is a subspace of $\text{row}_{\mathbb{F}_2}(\mathbf{A})$ and $W$ contains at least one element of $\{\mathbf{A}_1, \ldots, \mathbf{A}_{d^N}\}$. If $\dim W < \text{rank}_{\mathbb{F}_2}(\mathbf{A})$, then there are at least $d^{N-1}$ rows of $\mathbf{A}$ which are each linearly independent of $W$.*

The main message of the above is that as long as $W$ is "missing a direction" in $\mathbf{A}$'s rowspace, there are at least $1/d$ fraction of rows that would increase its dimensionality.

**Lemma 4.2.** *Let $Y_1, Y_2, \ldots$ where $Y_t \overset{iid}{\sim} \text{Unif}(\{\mathbf{A}_1, \ldots, \mathbf{A}_{d^N}\})$. We have that $m \lesssim (dN)^2 \log d$ samples suffice to ensure $\dim \text{span}\{Y_1, \ldots, Y_m\} = \text{rank}_{\mathbb{F}_2}(\mathbf{A})$ w.p. $\geq 2/3$.*

*Proof of Lemma 4.2.* Let $m$ be a positive integer. To each sequence $y = (\mathbf{y}_1, \ldots, \mathbf{y}_m) \in \{\mathbf{A}_1, \ldots, \mathbf{A}_{d^N}\}^m$ we associate another sequence $h^y = (h_1, \ldots, h_m)$ where $h_t^y := \text{span}\{\mathbf{y}_1, \ldots, \mathbf{y}_t\}$ is the cumulative span of the first $t$ vectors of the sequence $y$. Consider the directed graph $G = (V, E)$ in which

$$V := \{(\alpha, W) \mid \alpha \in [\text{rank}_{\mathbb{F}_2}(\mathbf{A})], W \text{ is a subspace of } \text{row}_{\mathbb{F}_2}(\mathbf{A})\}$$

The edgeset $E$ is defined as follows. For each $m \in \mathbb{N}_{>0}$ and each $y \in \{\mathbf{A}_1, \ldots, \mathbf{A}_{d^N}\}^m$, we include in $E$ the directed edge

$$\left((\dim h_t^y, h_t^y), (\dim h_{t+1}^y, h_{t+1}^y)\right) \in V \times V$$

for all $t \in [m-1]$, emphasizing that self-loops are allowed. Plainly stated, the construction places paths on the graph tracking the cumulative span and its dimension for every possible sequence. Notably, $V$ includes

the vector $v^* := (\mathrm{rank}_{\mathbb{F}_2}(\mathbf{A}), \mathrm{row}_{\mathbb{F}_2}(\mathbf{A}))$ because $\mathbf{A}$ trivially has a sequence of $\mathrm{rank}_{\mathbb{F}_2}(\mathbf{A})$ vectors whose span is $\mathrm{row}_{\mathbb{F}_2}(\mathbf{A})$. In particular, $v^*$ has only a single outgoing edge, which is also a self-loop.

It follows that the joint distribution of $Y_1, Y_2, \ldots$ defines a time-homogeneous Markov chain $\nu_1, \nu_2, \ldots$ on the state space $V$. For each $(v_1, \ldots, v_m) \in \mathrm{Paths}(G)$, denote $(\tilde{v}_1, \ldots, \tilde{v}_{\tilde{m}})$ as the truncated path obtained by removing self-loops, i.e. any vertex whose previous vertex is identical to itself, and $\tilde{v}_1 = v_1$. It follows that the probability $\dim \mathrm{span}\{Y_1, \ldots, Y_m\} \neq \mathrm{rank}_{\mathbb{F}_2}(\mathbf{A})$ is at most

$$
\begin{aligned}
\mathbb{P}(X_m \neq v^*) &= \sum_{(v_1, \ldots, v_m) \in \mathrm{Paths}(G): v_m \neq v^*} \mathbb{P}(X_1 = v_1, \ldots X_m = v_m) \\
&\stackrel{(a)}{=} \sum_{(v_1, \ldots, v_m) \in \mathrm{Paths}(G): v_k \neq v^*, \forall k \in [m]} \mathbb{P}(X_1 = v_1, \ldots X_m = v_m) \\
&\stackrel{(b)}{=} \sum_{(v_1, \ldots, v_m) \in \mathrm{Paths}(G): v_k \neq v^*, \forall k \in [m]} \mathbb{P}(X_1 = \tilde{v}_1, \ldots X_{\tilde{m}} = \tilde{v}_{\tilde{m}}) \times \prod_{k=1}^{m} \mathbb{P}(v_k \text{ is self-loop})^{\mathbf{1}[v_k \text{ is self-loop}]} \\
&\stackrel{(c)}{\leq} \sum_{(v_1, \ldots, v_m) \in \mathrm{Paths}(G): v_k \neq v^*, \forall k \in [m]} \left(1 - \frac{1}{d}\right)^{(\# \text{ self-loops in } (v_1, \ldots, v_m))} \\
&\stackrel{(d)}{\leq} \sum_{(v_1, \ldots, v_m) \in \mathrm{Paths}(G): v_k \neq v^*, \forall k \in [m]} e^{-\frac{m - \mathrm{rank}_{\mathbb{F}_2}(\mathbf{A}) + 1}{d}} \\
&\stackrel{(e)}{\leq} e^{-\frac{m - \mathrm{rank}_{\mathbb{F}_2}(\mathbf{A}) + 1}{d}} d^{N \, \mathrm{rank}_{\mathbb{F}_2}(\mathbf{A})} \\
&\stackrel{(f)}{\leq} 1/3
\end{aligned}
$$

where $(a)$ follows since $v^*$ is an absorbing state; $(b)$ follows by the Markov property and time-homogeneity; $(c)$ uses the observation that $\mathbb{P}(X_{t+1} = v \mid X_t = v) \leq 1 - d^{N-1}/d^N$ for $v \in V \setminus \{v^*\}$ by Lemma 4.1; $(d)$ uses the observation that there are $\geq m - \mathrm{rank}_{\mathbb{F}_2}(\mathbf{A}) + 1$ self-loops in paths never reaching $v^*$, otherwise $> (m-1) - (m - \mathrm{rank}_{\mathbb{F}_2}(\mathbf{A}) + 1) = \mathrm{rank}_{\mathbb{F}_2}(\mathbf{A}) - 2$ of the edges are associated with an increase of the cumulative span's dimension—implying the chain has to reach $v^*$; $(e)$ uses the observation that there are at most $\leq d^{N\tilde{m}}$ sequences in $\{\mathbf{A}_1, \ldots, \mathbf{A}_{d^N}\}^{\tilde{m}}$, each of which contributes to at most one length $\tilde{m}$ loop-less walk in the graph. Since the path mustn't terminate at $v^*$, $\tilde{m} \leq \mathrm{rank}_{\mathbb{F}_2}(\mathbf{A})$, from which the bound follows. Finally, $(f)$ follows from choosing $m = \mathrm{rank}_{\mathbb{F}_2}(\mathbf{A}) - 1 + \lceil d \log(3 d^{N \, \mathrm{rank}_{\mathbb{F}_2}(\mathbf{A})}) \rceil$, which is $O((dN^2)\log(d))$ since $\mathrm{rank}_{\mathbb{F}_2}(\mathbf{A}) = O(dN)$ (c.f. Lemma 2.1).

$\square$

Now, it just remains to verify Lemma 4.1.

*Proof of Lemma 4.1.* Consider the set of vectors

$$
\phi_i^n := [\mathbf{0}_d, \ldots, \mathbf{0}_d, \underbrace{\mathbf{e}_1 + \mathbf{e}_i}_{n^{th} \text{ position}}, \mathbf{0}_d, \ldots \mathbf{0}_d] \in \{0, 1\}^{dN}
$$

which are defined for all $i \in [d] \setminus \{1\}$ and $n \in [N]$. There must exist a $\phi_i^n$ which is linearly independent of $W$. Otherwise $W$ contains the subspace $\mathrm{row}_{\mathbb{F}_2}(\Phi)$ ($\Phi$ is defined in equation 11)—but $\mathrm{row}_{\mathbb{F}_2}(\Phi) = \mathrm{row}_{\mathbb{F}_2}(\mathbf{A})$ by Corollary 3.3, which contradicts the dimension of $W$. For this $\phi_i^n$, consider the set of unordered vector pairs $\{\{\mathbf{a}, \mathbf{b}\} \in \{\mathbf{A}_1, \ldots, \mathbf{A}_{d^N}\}^2 \mid \mathbf{a} + \mathbf{b} = \phi_i^n \mod 2\}$.

Notably, this set contains exactly $d^{N-1}$ pairs, as one must fix the column-block in the $n^{th}$ position to be $\mathbf{e}_1$ for one, which fixes the other to be $\mathbf{e}_i$—varying over the last $N - 1$ column-blocks with $d$ choices for each. For each pair, at least one of $\mathbf{a}$ or $\mathbf{b}$ must be linearly independent of $W$, for otherwise it contradicts that $\phi_i^n$ is not in the subspace $W$. Hence, one can find $\geq d^{N-1}$ rows of $\mathbf{A}$, each individually linearly independent of $W$.

$\square$

In the next section, we prove the correctness of Algorithm 1.

## 5 Proof of Theorem 1.3

Let $m$ denote the quantity in Lemma 4.2 of the previous section. The algorithm asserted in Theorem 1.3 simply draws $m' := \max\{m, dN\} = O((dN)^2 \log d)$ samples, takes $S$ as the set of associated indices, and runs Algorithm 1 for this input $S$.

The runtime is immediate. Indeed, the systems in steps (1) and (2) are size $m' \times dN$ and consistent, so Gauss-Jordan on each terminates in time $O(m'(dN)^2)$. The algorithm takes an additional $O(N)$ time per queried entry in step (3), thus $O(qN + m'(dN)^2)$ overall.

We now turn towards establishing the correctness of Algorithm 1. As our choice of $m'$ satisfies Lemma 2.2, we have that $\mathrm{row}_{\mathbb{F}_2}(\mathbf{D}_S\mathbf{A}) = \mathrm{row}_{\mathbb{F}_2}(\mathbf{A})$ and $\mathrm{row}_{\mathbb{R}}(\mathbf{D}_S\mathbf{A}) = \mathrm{row}_{\mathbb{R}}(\mathbf{A})$ holds w.p. $\geq 2/3$. Thus, it suffices to prove the following.

**Lemma 5.1.** *Let $\mathbf{U} \in \otimes_{i=1}^{N} \mathbb{R}_{\neq 0}^d$ be a rank-1 tensor. Assume the input $S$ satisfies both*

$$\mathrm{row}_{\mathbb{F}_2}(\mathbf{D}_S\mathbf{A}) = \mathrm{row}_{\mathbb{F}_2}(\mathbf{A}), \quad \mathrm{row}_{\mathbb{R}}(\mathbf{D}_S\mathbf{A}) = \mathrm{row}_{\mathbb{R}}(\mathbf{A}).$$

*Then, the output of Algorithm 1 satisfies $\hat{\mathbf{U}} = \mathbf{U}$.*

To start, we establish a useful helper lemma.

**Lemma 5.2.** *Suppose $\mathbf{A}\mathbf{x} = \mathbf{b}$ is a consistent linear system over a field $\mathbb{K}$, for which*

$$\mathbf{A} = \begin{pmatrix} \mathbf{A}_1 \\ \mathbf{A}_2 \end{pmatrix}, \quad \mathbf{b} = \begin{pmatrix} \mathbf{b}_1 \\ \mathbf{b}_2 \end{pmatrix}.$$

*Suppose $\mathrm{row}_{\mathbb{K}}(\mathbf{A}_1) = \mathrm{row}_{\mathbb{K}}(\mathbf{A})$. If $\mathbf{x}^*$ satisfies $\mathbf{A}_1\mathbf{x}^* = \mathbf{b}_1$, then $\mathbf{A}_2\mathbf{x}^* = \mathbf{b}_2$ holds (and evidently $\mathbf{A}\mathbf{x}^* = \mathbf{b}$).*

*Proof of Lemma 5.2.* By consistency there exists a $\mathbf{z}$ such that $\mathbf{A}_1\mathbf{z} = \mathbf{b}_1$ and $\mathbf{A}_2\mathbf{z} = \mathbf{b}_2$. Since $\mathbf{x}^*$ satisfies $\mathbf{A}_1\mathbf{x}^* = \mathbf{b}_1$, we have $\mathbf{x}^* - \mathbf{z} \in \ker_{\mathbb{K}}(\mathbf{A}_1) = \mathrm{row}_{\mathbb{K}}(\mathbf{A}_1)^{\perp} = \mathrm{row}_{\mathbb{K}}(\mathbf{A})^{\perp} = \ker_{\mathbb{K}}(\mathbf{A})$. Hence, $\mathbf{A}(\mathbf{x}^* - \mathbf{z}) = \mathbf{0}$, i.e. $\mathbf{A}\mathbf{x}^* = \mathbf{A}\mathbf{z}$ implying $\mathbf{A}_2\mathbf{x}^* = \mathbf{A}_2\mathbf{z} = \mathbf{b}_2$. $\quad\square$

Now, we have all the tools to prove Lemma 5.1.

*Proof of Lemma 5.1.* From equation 8, let $\mathbf{P}_S \in \{0,1\}^{d^N \times d^N}$ be a permutation matrix such that

$$\mathbf{P}_S\mathbf{A} = \begin{pmatrix} \mathbf{D}_S\mathbf{A} \\ \mathbf{D}_{\bar{S}}\mathbf{A} \end{pmatrix}, \quad \mathbf{P}_S f(\mathbf{U}) = \begin{pmatrix} \mathbf{D}_S f(\mathbf{U}) \\ \mathbf{D}_{\bar{S}} f(\mathbf{U}) \end{pmatrix}, \quad \mathbf{P}_S \tilde{f}(\mathbf{U}) = \begin{pmatrix} \mathbf{D}_S \tilde{f}(\mathbf{U}) \\ \mathbf{D}_{\bar{S}} \tilde{f}(\mathbf{U}) \end{pmatrix}. \tag{12}$$

By Observation 1 both systems $[\,\mathbf{D}_S\mathbf{A} \mid \mathbf{D}_S f(\mathbf{U})\,]_{\mathbb{F}_2}$ and $[\,\mathbf{D}_S\mathbf{A} \mid \mathbf{D}_S \tilde{f}(\mathbf{U})\,]_{\mathbb{R}}$ are consistent—hence steps (1) and (2) return a $\mathbf{y}_1$ and $\mathbf{y}_2$ for which $\mathbf{D}_S\mathbf{A}\mathbf{y}_1 = \mathbf{D}_S f(\mathbf{U})$ and $\mathbf{D}_S\mathbf{A}\mathbf{y}_2 = \mathbf{D}_S \tilde{f}(\mathbf{U})$.

We invoke Lemma 5.2 to the two systems induced by equation 12. Specifically, we assign $\mathbf{A} \leftarrow \mathbf{P}_S\mathbf{A}$, $\mathbf{b} \leftarrow \mathbf{P}_S f(\mathbf{U})$ and use that $\mathbf{D}_S\mathbf{A}\mathbf{y}_1 = \mathbf{D}_S f(\mathbf{U})$, upon which the lemma lets us conclude $\mathbf{P}_S\mathbf{A}\mathbf{y}_1 = \mathbf{P}_S f(\mathbf{U})$. Had we instead taken $\mathbf{b} \leftarrow \mathbf{P}_S \tilde{f}(\mathbf{U})$ and used $\mathbf{D}_S\mathbf{A}\mathbf{y}_2 = \mathbf{D}_S \tilde{f}(\mathbf{U})$, we would conclude $\mathbf{P}_S\mathbf{A}\mathbf{y}_2 = \mathbf{P}_S \tilde{f}(\mathbf{U})$. Since $\mathbf{P}_S$ is a row permutation, this implies

$$\mathbf{A}\mathbf{y}_1 = f(\mathbf{U}), \quad \mathbf{A}\mathbf{y}_2 = \tilde{f}(\mathbf{U}).$$

Hence, recalling the map $\varphi$ and the definition of $f$ and $\tilde{f}$,

$$\varphi^{-1}(\mathbf{A}\mathbf{y}_1) = (\mathrm{sign} \circ \mathrm{vec}_{\pi})(\mathbf{U})$$
$$\exp(\mathbf{A}\mathbf{y}_2) = (\mathrm{abs} \circ \mathrm{vec}_{\pi})(\mathbf{U}).$$

Evidently,

$$\mathbf{U}_{(i_1,\ldots,i_N)} = \varphi^{-1}(\langle \mathbf{A}_{\pi(i_1,\ldots,i_N)}, \mathbf{y}_1 \rangle) \exp\left(\langle \mathbf{A}_{\pi(i_1,\ldots,i_N)}, \mathbf{y}_2 \rangle\right) = \hat{\mathbf{U}}_{(i_1,\ldots,i_N)},$$

which is what we wanted to show. $\quad\square$

## 6 Proof of Theorem 1.5, Corollary 1.6

As we'll describe, the lower bound in Theorem 1.5 is a consequence of the following.

**Lemma 6.1.** *Consider a variant of the coupon collector problem in which there are $N \in \mathbb{N}_{>0}$ urns, each containing $d \in \mathbb{N}_{>0}$ unique balls. Suppose each draw is given by a uniformly random choice of one ball from each and every urn, for a total of $N$ balls per draw.*

*There exists absolute constants $n_0 \in \mathbb{N}$ and $C > 0$ such that $dN \geq n_0$ implies that if less than $Cd \log dN$ draws are taken, then at least one ball is missed w.p. $> 2/3$.*

By the classic variant, it is easy to see $\Omega(d \log d)$ is necessary. Since our main result strives for optimal $d$ dependence, this would be enough. For this reason we leave the proof to Appendix A, which involves recursively applying Hoeffding's lemma to a particular martingale sequence. However, we feel this result clarifies the "coupon collector effect" frequently referred to in the tensor completion literature—often as a remark used to justify the presence of logarithmic factors in the upper bounds. In contrast, our lower bound explicitly uses such an argument. We now detail the lower bound's proof.

*Proof of Theorem 1.5.* Let $\mathbf{u}_1, \ldots, \mathbf{u}_N \overset{iid}{\sim} \mathrm{Unif}(\{\pm \sigma^{\frac{1}{N}}\}^d)$ and let the random tensor $\mathbf{U}$ be given by $\mathbf{U} := \mathbf{u}_1 \otimes \cdots \otimes \mathbf{u}_N$. Recalling each entry of $\mathbf{U}$ is dependent on $N$ out of $dN$ variables (c.f. equation 1), we say a sampled entry $\mathbf{U}_{(i_1, \ldots, i_N)}$ *collects* the variable $(\mathbf{u}_k)_\ell$ if the former is dependent on the latter. We can correspond the samples with the coupon collecting procedure in Lemma 6.1. Concretely, we assign each $\mathbf{u}_i$ to the urn $i$, and each coordinate variable $(\mathbf{u}_i)_j$ with the $j^{th}$ ball in the $i^{th}$ urn. Let $m$ denote the quantity indicated by (the proof of) Lemma 6.1. Supposing the algorithm collects less than $m$ samples, i.e. $|S| < m$, then by Lemma 6.1 the algorithm won't *collect* some variable $(\mathbf{u}_{k'})_{\ell'}$ w.p. $> 2/3$. We condition on this event for the rest of the proof.

Denote $\mathbf{U}^{k'\ell'}(\omega)$ as $\mathbf{U}$ conditioned on the assignment of all the variables in $\mathbf{u}_1, \ldots, \mathbf{u}_N$ *except* $(\mathbf{u}_{k'})_{\ell'}$ to values specified by outcome $\omega$. Similarly, let $\mathbf{U}_+^{k'\ell'}(\omega)$ and $\mathbf{U}_-^{k'\ell'}(\omega)$ denote the nonrandom tensor obtained from these by then fixing $(\mathbf{u}_{k'})_{\ell'}$ to $+\sigma^{\frac{1}{N}}$ and $-\sigma^{\frac{1}{N}}$, respectively. For each $\omega$, $\mathbf{U}_+^{k'\ell'}(\omega)$ and $\mathbf{U}_-^{k'\ell'}(\omega)$ are vectors in the inner product space $(\otimes_{i=1}^N \mathbb{R}^d, \|\cdot\|_F)$. Hence, they are bisected by the hyperplane $\langle \mathbf{T}, \mathbf{U}_+^{k'\ell'}(\omega) - \mathbf{U}_-^{k'\ell'}(\omega) \rangle = 0$, separated by distance

$$\|\mathbf{U}_+^{k'\ell'}(\omega) - \mathbf{U}_-^{k'\ell'}(\omega)\|_F = \|(2\sigma^{\frac{1}{N}} \mathbf{e}_{\ell'}) \otimes (\otimes_{i \neq k'} \mathbf{u}_i)\|_F = |2\sigma^{\frac{1}{N}}| \left(\sqrt{\sigma^{\frac{2}{N}} d}\right)^{N-1} = 2\sigma\sqrt{d^{N-1}}.$$

Fixing the outcomes of $S$ and $B$ (where $S$ outcomes are restricted to the aforementioned event), we have that the algorithm output is constant under outcomes $\mathbf{U}_+^{k'\ell'}(\omega)$ and $\mathbf{U}_-^{k'\ell'}(\omega)$, and must lie on one side of hyperplane. However, w.p. $\geq 1/2$ the target tensor is on the other side of the hyperplane, incurring error $\geq \sigma\sqrt{d^{N-1}}$ w.p. $> (2/3) \cdot (1/2) = 1/3$. $\square$

Under the following assumption, we now prove Corollary 1.6 by approximating the support of the above distribution by tensors of rank $\gtrsim r$. Recall this property essentially asserts the estimator's error is weakly increasing for perturbations to its input.

**Assumption 6.2.** *Let $\mathcal{D}, \mathcal{D}'$ be two distributions over $\otimes_{i=1}^N \mathbb{R}^d$. The randomized estimator $\mathcal{A}(\cdot, S, B)$ satisfies a.s. (almost surely) over randomness in $S, B$, and draws $\mathbf{U} \sim \mathcal{D}, \mathbf{T} \sim \mathcal{D}'$*

$$\|\mathcal{A}(\mathbf{U}, S, B) - \mathbf{U}\|_F \leq \|\mathcal{A}(\mathbf{U} + \mathbf{T}, S, B) - \mathbf{U}\|_F.$$

*Proof of Corollary 1.6.* We fix an arbitrary $\epsilon \in (0, \sigma)$. Consider a Hadamard basis over dimension $d$. From this basis we may obtain a set of $d^N$ mutually orthogonal rank-1 tensors with entries in $\pm 1$, say $\mathcal{T}$. We let $\mathbf{V}_1, \ldots, \mathbf{V}_r$ denote an arbitrary but fixed $r$-size subset of $\mathcal{T}$.

Let $\mathbf{u}_1, \ldots, \mathbf{u}_N \overset{iid}{\sim} \mathrm{Unif}(\{\pm (\sigma - \epsilon)^{1/N}\}^d)$ and let the random tensor $\mathbf{V}_0$ be given by $\mathbf{V}_0 := \mathbf{u}_1 \otimes \cdots \otimes \mathbf{u}_N$.

For each $K \in \mathbb{R}_{>0}$ and outcome $\omega$, we define $\mathbf{U}_K(\omega) := \mathbf{V}_0(\omega) + \frac{1}{K} \sum_{i=1}^{r} \mathbf{V}_i$. Let $\mathcal{D}_K^*$ be the distribution of the random variable $\mathbf{U}_K$. Notably, for $K \geq r/\epsilon$ we have a.s. that $\|\mathbf{U}_K\|_\infty \leq \sigma$ and $\mathbf{U}_K$ is at least rank-$(r-1)$, due to possible cancellations.

Suppose by contradiction that $\mathcal{A}$ is a randomized estimator where $m$ is below the threshold in Theorem 1.5 but w.p. $> 2/3$ we have that $\|\mathcal{A}(\mathbf{U}_K, S, B) - \mathbf{U}_K\|_F < (1/2)\sigma\sqrt{d^{N-1}}$, where $S$ and $B$ are as in definition 1.4. By reverse triangle inequality, $\|\mathcal{A}(\mathbf{U}_K, S, B) - \mathbf{V}_0\|_F - \|\mathbf{V}_0 - \mathbf{U}_K\|_F \leq (1/2)\sigma\sqrt{d^{N-1}}$, implying

$$\|\mathcal{A}(\mathbf{U}_K, S, B) - \mathbf{V}_0\|_F \leq (1/2)\sigma\sqrt{d^{N-1}} + \frac{1}{K}\|\sum_{i=1}^{r} \mathbf{V}_i\|_F$$
$$= (1/2)\sigma\sqrt{d^{N-1}} + \frac{r}{K}\|\mathbf{V}_1\|_F$$
$$= (1/2)\sigma\sqrt{d^{N-1}} + \frac{r}{K}\sqrt{d^N}$$

Taking $K > 2r\sqrt{d}/\sigma$, we have a contradiction to Theorem 1.5 since w.p. $> 2/3$

$$\|\mathcal{A}(\mathbf{V}_0, S, B) - \mathbf{V}_0\|_F \overset{6.2}{\leq} \|\mathcal{A}(\mathbf{U}_K, S, B) - \mathbf{V}_0\|_F < \sigma^N\sqrt{d^{N-1}}.$$

$\square$

# 7   Conclusion and Open Questions

This paper presents a novel analysis of the rank-1 tensor completion problem, which recasts it in terms of a special pair of linear systems and leverages this viewpoint to improve upon the previously established sample complexity bounds. When $N \asymp 1$, we prove that $O(d^2 \log d)$ uniformly observed entries are sufficient to exactly recover a rank-1 tensor with nonzero entries (c.f. Theorem 1.3), while $\Omega(d \log d)$ samples are necessary (c.f. Theorem 1.5), even for higher-rank tensors (c.f. Corollary 1.6). Notably, neither quantity depends on the incoherence $\mu$.

One of our main challenges involves a novel matrix sketch problem over $\mathbb{F}_2$, leading to the sample complexity upper bound's quadratic dependence on $d$. As asserted at the end of Section 2, we conjecture the upper bound can be improved to $O(d \log d)$ to match the lower bound, primarily by refining Lemma 4.1. This is motivated by the observation that the algorithm is optimal in $d$ when the input tensor $\mathbf{U}$ is unsigned, as detailed at the end of Section 2. We leave the resolution of this conjecture to the future work.

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

## A    Appendix A : Proof of Lemma 6.1

In this section, we prove Lemma 6.1.

- Let $Z_t^i$ for $i \in [N]$ denote the random variable counting the remaining balls in urn $i$ yet to be seen by and including the $t^{th}$ draw.

- Let $I_t^i$ for $i \in [N]$ denote the indicator random variable which is '1' if the $t^{th}$ draw sees a previously unseen ball in urn $i$, and is '0' otherwise.

- Let $\mathcal{F}_t$ denote the natural filtration generated by the random variables $\{Z_1^i, Z_2^i, \ldots, Z_t^i\}_{i=1}^N$.

Finally, denote $Z_t := \sum_{i=1}^N Z_t^i$ and $I_t := \sum_{i=1} I_t^i$. In what follows, it is helpful to note that $Z_{t+1} = Z_t - I_{t+1}$.

**Lemma A.1.** *For any $s > 0$, $d \geq 2$, and $t \in \mathbb{N}_{>0}$, we have*

$$\mathbb{E}(e^{-sZ_{t+1}}) \leq e^{\frac{s^2 dN}{8} - s(1-\frac{1}{d})^t \frac{dN}{2}} \tag{13}$$

*Proof.* Denoting $\alpha := 1 - \frac{1}{d}$,

$$
\begin{aligned}
\mathbb{E}(e^{-sZ_{t+1}}) &= \mathbb{E}(\mathbb{E}(e^{-s(Z_t - I_{t+1})} \mid \mathcal{F}_t)) \\
&= \mathbb{E}(e^{-sZ_t}\mathbb{E}(e^{sI_{t+1}} \mid \mathcal{F}_t)) \\
&\overset{(a)}{=} \mathbb{E}(e^{-sZ_t}\prod_{i=1}^N \mathbb{E}(e^{sI_{t+1}^i} \mid \mathcal{F}_t)) \\
&\overset{(b)}{\leq} \mathbb{E}\left(e^{-sZ_t}\prod_{i=1}^N e^{s\mathbb{E}(I_{t+1}^i \mid \mathcal{F}_t)) + \frac{s^2}{8}}\right) \\
&\overset{(c)}{=} e^{\frac{s^2 N}{8}}\mathbb{E}\left(e^{-sZ_t}\prod_{i=1}^N e^{s\frac{Z_t^i}{d}}\right) \\
&= e^{\frac{s^2 N}{8}}\mathbb{E}\left(e^{-sZ_t}e^{s\frac{Z_t}{d}}\right) \\
&= e^{\frac{s^2 N}{8}}\mathbb{E}\left(e^{-s\alpha Z_t}\right) \\
&\overset{(d)}{\leq} e^{\frac{s^2 N}{8}}\left(e^{\frac{\alpha^2 s^2 N}{8}}\mathbb{E}\left(e^{-s\alpha^2 Z_{t-1}}\right)\right) \\
&\leq \ldots \\
&\leq e^{\frac{s^2 N}{8}(1+\alpha^2+\alpha^4+\cdots+\alpha^{2(t-1)})}\mathbb{E}\left(e^{-s\alpha^t Z_1}\right) \\
&\overset{(e)}{=} e^{\frac{s^2 N}{8}(1+\alpha^2+\alpha^4+\cdots+\alpha^{2(t-1)})}e^{-s\alpha^t(d-1)N} \\
&\leq e^{\frac{s^2 N}{8}(1+\alpha+\alpha^2+\cdots+\alpha^{(t-1)})}e^{-s\alpha^t(d-1)N} \\
&\leq e^{\frac{s^2 dN}{8} - s\alpha^t\frac{dN}{2}}
\end{aligned}
$$

where $(a)$ follows since each urn is sampled from independently; $(b)$ applies Hoeffding's lemma; $(c)$ uses the observation that, given the filtration up to time $t$, the $i^{th}$ urn at time $t+1$ "sees" a new ball if sampling one of $Z_t^i$ uncollected balls out of $d$; $(d)$ is the first recursive application of the bound; and $(e)$ uses the simple observation that the first draw always "sees" $N$ balls, so a.s. $Z_1 = (d-1)N$.

$\square$

We are now in a position to prove Lemma 6.1.

*Proof of Lemma 6.1.* Fix $\beta \in (0,1)$ to be decided later. Assume that the following holds

$$t \leq \beta d \log dN. \tag{14}$$

We have $(1 - \frac{1}{d})^t \gtrsim e^{-\frac{t}{d}} \geq (dN)^{-\beta}$, so that for any $s > 0$ and $\epsilon > 0$, by Lemma A.1,

$$\mathbb{P}(Z_{t+1} \leq \epsilon) \leq e^{s\epsilon} \mathbb{E}(e^{-sZ_{t+1}}) \leq e^{s\epsilon + \frac{s^2 dN}{8} - s(1 - \frac{1}{d})^t \frac{dN}{2}} \lesssim e^{s\epsilon + \frac{s^2 dN}{8} - \frac{s}{2}(dN)^{1-\beta}}.$$

Let us constrain $\epsilon \in (0, \frac{1}{2}(dN)^{1-\beta})$ and take $s = \frac{4}{dN}(\frac{1}{2}(dN)^{1-\beta} - \epsilon)$ to give

$$\mathbb{P}(Z_{t+1} \leq \epsilon) \lesssim \exp\left(-\frac{2}{dN}\left(\frac{1}{2}(dN)^{1-\beta} - \epsilon\right)^2\right).$$

Suppose $\epsilon = \frac{1}{4}(dN)^{1-\beta}$ and $dN \geq 7$ so that

$$\mathbb{P}(Z_{t+1} \leq \frac{1}{4}(dN)^{1-\beta}) \leq \exp\left(-\frac{(dN)^{1-2\beta}}{8}\right).$$

In particular, for $\beta = 1/4$ and $dN \geq 78$, this implies

$$\mathbb{P}(Z_{t+1} \leq 1) \leq \mathbb{P}(Z_{t+1} \leq \frac{1}{4}(dN)^{3/4}) \leq \exp\left(-\frac{\sqrt{dN}}{8}\right) < \frac{1}{3}.$$

Thus, $\mathbb{P}(Z_{t+1} > 1) > \frac{2}{3}$. In other words, w.p. $> 2/3$ there remains an unseen ball if equation 14 holds for $\beta = 1/4$ and $dN \geq 78$. $\qquad\square$

## B   Appendix B: $\tilde{\Phi}$ for Proof of Lemma 2.1

In establishing the $\mathbb{R}$ rank of $\mathbf{A}$, the following matrix is referred to.

