# OpenReview forum: "Simple and Nearly-Optimal Sampling for Rank-1 Tensor Completion via Gauss-Jordan"
_TMLR — Accepted by TMLR_

### Review · Reviewer_VEAs · 2025-07-06

**Summary Of Contributions:**

This paper studies the problem of tensor-completion under low rank constraints. Specifically, the paper assumes that the true tensor is of rank-1 and the aim is to recover the true tensor upon the observation of some uniformly sampled entries. The main contribution is to propose an efficient and simple algorithm with less sample complexity and computational complexity. Specifically, the paper shows that it requires $m=O(d^2\log d)$ samples and run $O(md^2)$ to recover the tensor with probability $\geq2/3$, where $d$ is the dimension. The basic idea is to reduce the problem to two linear system solvers, one on the field $\mathbb{F}_2$ and one one $\mathbb{R}$. These two systems can be efficiently addressed by the Gauss-Jordan method.

**Audience:**

Yes

**Broader Impact Concerns:**

I do not see broader impact concerns.

**Claims And Evidence:**

Yes

**Requested Changes:**

- The paper proposed an interesting algorithm, and gave some theoretical analysis. It is interesting to include an empirical verification of the proposed algorithm, and do some experimental comparison with existing methods to show that it really works in practice.

- What is the challenge to extend the discussion from rank-1 problem to general low-rank problems. It would be beneficial to add some discussions. Furthermore, the comparison of the sample complexity and computational complexity is on the general low-rank problems of existing methods. How does the proposed algorithm compare to existing studies of the rank-1 problem in terms of sample and computational complexity?

- It would be helpful to add more explanations on the derivation of the algorithms. Some notations should be explained clearly. For example, people in applied field may not be familiar with abstract algebra. Therefore, the meaning of $\mathbb{F}$ should be given clearly

- There are many typos in the paper. Below are some examples.

Remark 1.6: "exists distinct tensors"

Above Eq (6): "this is can"

Below Eq (6): "rows adheres"

bottom of Page 5: "conditions holds"

Section 5: 'lets $S$"

Lemma 6.1: "There exists absolute constants"

**Strengths And Weaknesses:**

**Strength**

- The main strength of the submission is to propose an efficient algorithm which achieves much less computational complexity and much less sample complexity. The existing algorithms have complexity with an exponential dependency on the incoherence. The paper shows that this dependency can be removed.

- The proposed algorithm is simple to implement as it only requires to solve two linear systems of equations. As a comparison, the existing algorithms are much more complicated as they invoke tools from algebraic geometry and matroid theory.

**Weakness**
- The paper only considers the rank-1 tensor completion problems. This limits the application domain of the proposed algorithms as the practical problems often have low-rank structures with larger ranks.
- It seems that the paper only considers the noiseless case. It is not clear to me whether the proposed algorithm can still enjoy the improved sample and computational complexity if there are noises in the system.
- The submission is too technical and difficult to follow as it introduces mathematically-heavy deductions without enough explanations in the derivation of the algorithm. It would be better to introduce more explanations.
- The paper does not contain empirical verifications of the proposed algorithms. Therefore, it is not clear how the proposed algorithm compares with existing methods.

---

> ### Author Response · Authors · 2025-07-30
> **Response to Reviewer VEAs**
>
> Comment: "The paper proposed an interesting algorithm, and gave some theoretical analysis. It is interesting to include an empirical verification of the proposed algorithm, and do some experimental comparison with existing methods to show that it really works in practice."
>
> Response: We feel that the algorithm is not the focus of this work, rather it serves as a means to achieve a novel mathematical analysis of the tensor completion problem.
> Moreover, in the "Introduction" section we've noted an assumption on infinite precision arithmetic.
> For these reasons, we prefer to keep the paper as is. However, if other reviewers agree with this request, then we would be happy to add some numerical demonstration.
>
> ==========================
>
> Comment: "What is the challenge to extend the discussion from rank-1 problem to general low-rank problems. It would be beneficial to add some discussions. Furthermore, the comparison of the sample complexity and computational complexity is on the general low-rank problems of existing methods. How does the proposed algorithm compare to existing studies of the rank-1 problem in terms of sample and computational complexity?"
>
> Response: To your first concern, higher rank tensors introduce additional sums that cannot be incorporated trivially into the described linear systems.
> We added a clarification of this point in "Our Results", following the first theorem.
> To your second concern, we'd like to clarify that the comparison of sample complexity is actually on the rank-1 performance of existing methods, i.e. the described results have been reported for the case of rank-1.
>
> ==========================
>
> Comment: "It would be helpful to add more explanations on the derivation of the algorithms. Some notations should be explained clearly. For example, people in applied field may not be familiar with abstract algebra. Therefore, the meaning of $\mathbb{F}$ should be given clearly."
>
> Response: We've since augmented the high-level explanations of our main results, and took care in avoiding nonstandard notation prior to the "Notation" paragraph.
>
> ==========================
>
> Comment: "here are many typos in the paper."
>
> Response: Thank you for pointing these out. The listed 2-4 are genuine typos, and we've since fixed these.
> The rest we've modified for clarity.
> Moreover, we've thoroughly parsed the manuscript and corrected any remaining typos, also marked in blue.
>
> ==========================

---

> > ### Comment · Reviewer_VEAs · 2025-08-07
> > **Noise case**
> >
> > Dear Authors,
> >
> > Thank you for preparing the revision according to the comments. I also mentioned that the paper seems to consider only the noiseless case. It is not clear to me whether the proposed algorithm can still enjoy the improved sample and computational complexity if there are noises in the system. I am wondering if the authors can share their idea regarding this comment. Thank you.
> >
> > Best regards,

---

> > > ### Author Response · Authors · 2025-08-11
> > > **Noise Case Response**
> > >
> > > Thank you for the insightful observation. The proposed algorithm is not suitable for the noisy case, the limitation being that even mild additive noise can increase the effective rank of a tensor. And because the characterization does not trivially extend to higher-rank tensors, this makes the analysis inapplicable as-is.
> > >
> > > This limitation also partly explains our minimal focus on practical applications: the current approach is largely restricted to the noiseless, rank-1 setting, and thus has limited utility in real-world applications. Therefore, our primary motivation is to advance a novel theoretical understanding of the tensor completion problem from an information-theoretic perspective, rather than propose an immediately deployable algorithm.
> > >
> > > That being said, we believe the current framework may offer a useful primitive for future extensions to noisy and higher-rank tensors. We hope that by sharing our findings in this simplified regime, the work can serve as a useful stepping stone for further theoretical and practical developments in tensor recovery.

---

> > > > ### Comment · Reviewer_VEAs · 2025-08-12
> > > >
> > > > Thank you for your responses, which clarify my thoughts.

---

### Review · Reviewer_EHqm · 2025-07-14

**Summary Of Contributions:**

This paper studies the rank-1 tensor completion problem. In fact, the problem is to recover the entirety of a rank-1 tensor observing only a small subset of its entries. The authors provide an algorithm to do so boiling down to solving a pair of random linear systems. They show sample and time complexity bounds for it. Moreover, they show a sample complexity lower bound for a broader class of algorithms, highlighting that their method is close-to-optimal.

**Audience:**

Yes

**Broader Impact Concerns:**

No concern.

**Claims And Evidence:**

Yes

**Requested Changes:**

I am not familiar with this literature, however it would be nice to add some experiments, either supporting the theoretical claims or using the proposed algorithm on a real-world application.

Moreover, regarding the presentation of the paper I would suggest:
1. adding some real-world application examples of the rank-1 tensor completion problem in the introduction of the paper;
2. I would not introduce the theorems 1.3, 1.5 and corollary 1.7 in the introduction section. I would use it to present the problem, discuss the prior work and present the contributions concisely and clearly at the end of it (as well as the notations eventually);
3. adding a conclusion at the end of the paper.

**Strengths And Weaknesses:**

Strengths

1. All theoretical claims are proved.
2. Related works are discusses.
3. The tightness of the bound is discussed, even if no intuition about the factor $d$ mismatch between the upper and lower bounds is provided.

Weaknesses

1. No real-world application examples of the rank-1 tensor completion problem are provided.
2. No experiment corroborating the theoretical results nor real-worlds experiment is provided.
3. In my opinion, the presentation of the paper lacks clarity.

---

> ### Author Response · Authors · 2025-07-30
> **Response to Reviewer EHqm**
>
> Comment: "adding some real-world application examples of the rank-1 tensor completion problem in the introduction of the paper;"
>
> Response: Of course, in any real-world system well-approximated as a rank-1 tensor the proposed approach is applicable, but we understand your question as to if there are any systems in which they are canonically modeled as such.
> Unfortunately the previous literature for rank-1 tensor completion lacks explicit applications, instead focusing on the theoretical and fundamental aspects of the more complicated higher rank problem.
> However, we should also mention that if the tensor has only binary entries, then this is highly related to the canonical k-XOR-SAT problem studied in the complexity theory community, since such a tensor is expressable as a linear system over $\mathbb{F}_2$, as discussed in our work.
> This connection is also made in Stephan $\&$ Zhu (2024), but as the focus of the work is not applications and this connection is well-known, we did not feel it was appropriate for our submission.
>
> ==========================
>
> Comment: "I would not introduce the theorems 1.3, 1.5 and corollary 1.7 in the introduction section. I would use it to present the problem, discuss the prior work and present the contributions concisely and clearly at the end of it (as well as the notations eventually);"
>
> Response: Agreed. We have reorganized the "Introduction" section to accommodate your suggestion.
>
> ==========================
>
> Comment: "adding a conclusion at the end of the paper."
>
> Response: We have modified the "Open Questions" section to "Conclusion and Open Questions", which now includes a thorough, but concise summary of our contributions and possible future directions.

---

### Review · Reviewer_WGnL · 2025-07-16

**Summary Of Contributions:**

This paper achieves near-optimal sample complexity for rank-1 tensor completion by introducing a novel linear algebraic characterisation that decomposes the problem into two coupled linear systems over $\mathbb{F}_2$ and $\mathbb{R}$. The resulting algorithm requires only $O(d^2 \log d)$ samples with $O(md^2)$ runtime—a significant improvement over existing sample complexity upper bounds of $d^{1.5} \mu^{\Omega(1)} \log^{\Omega(1)} d$, where $\mu$ is the incoherence parameter. The presented upper bound has no dependence on $\mu$, revealing a complexity gap between rank-1 and higher-rank problems. The authors calculate the computational complexity of the algorithm via Gauss-Jordan. The authors also provide an $\Omega(d \log d)$ lower bound for the sample complexity against broad classes of algorithms.

**Audience:**

Yes

**Broader Impact Concerns:**

N.A.

**Claims And Evidence:**

Yes

**Requested Changes:**

Please address the weaknesses described above.

**Strengths And Weaknesses:**

**Strengths:**

The presented problem reformulation converts the tensor completion problem into solving two linear systems:
1. A binary system over $\mathbb{F}_2$ (for signs)
2. A real system over $\mathbb{R}$ (for magnitudes)
This separation makes the problem more tractable algorithmically.

**Weaknesses:**

The paper reads like a collection of small, fragmented definitions and results, and lacks coherent flow and organisation. I would strongly recommend that the paper be rewritten before submission for publication. In particular, various environments including “Theorem”, “Definition”, “Fact”, “Claim” are overused.

By “nearly-optimal” do you mean the sample complexity is near-optimal in d? However your upper and lower bounds are factor of d apart. I wonder how this can be used to support the near-optimality claim.

Introduction: “several applications” can you elaborate?

I understand that $\otimes_{i=1}^N \mathbb{R}^d_{\ne 0}$ means you want to exclude the all-zero vector, but I’m unsure how universal this notation is. Better define notation here.

Theorem 1.1 seems unclear to me: why is it an informal version of *three* results simultaneously? Maybe it suffices to describe the theorem in text rather than using the Theorem environment if you are being informal here. What do you mean by “solves rank-1 completion”? I guess you mean by recovering each entry of U correctly with probability at least 2/3? But why failure probability is used later on instead as a measure of the quality of recovery? Also the sample size m should be defined earlier when rank-1 completion is defined. Use $m\gtrsim d\log d$ in the last sentence. I think it’s better to call the problem “rank-1 tensor completion” throughout.

below Theorem 1.1: incoherence $\mu$ needs to be defined and references where the definition is drawn from need to be added.

Definition 1.2: what does CPD stand for?

Footnotes are overused in the paper, and they also appear wordy and unclear, please streamline them. Footnote 1-2: what is BPP? why 2/3 rather than $1-\delta$ for an arbitrarily small constant $\delta$? The four footnotes on p.2 can be largely compressed or stated in text IMO. Footnote 7 can also be absorbed into the Notation paragraph. Footnote 8 should also be defined inline. Also make the writing more formal, avoiding colloquial phrases like “blowup” or “the comparison is not that imprecise….”.

This paper uses extensive notation, so the Notation paragraph on p.3 should be moved much earlier to help readers.

Theorem 1.3: Why introduce q here rather than directly referencing the uniform sampling probability? Also, again, how is the precision of tensor completion defined in this theorem?

Above Definition 1.4: “We believe this is an artifact of our upper bound”. Is there any reason/ related references to back this claim? If not, it is possible the derived lower bound is loose.

Definition 1.4: The joint entropy here does not make sense. Shouldn’t it be the conditional entropy H(A|U,S,B) = 0? meaning Given knowledge of U, S, and B, there is no uncertainty about A's output?

Theorem 1.5: I do not understand this lower bound here. What happens with probability 2/3? Since this is a lower bound ie hardness result, the reader would expect that it holds with high probability. Also in text following this theorem, Assumption 6.3 hasnt been stated at this point, which hinders the readability of the results.

Corollary 1.7: can you define what a Hadamard matrix of order d is?

While I have not verified the proofs in detail, they appear to rely on elementary probabilistic and linear algebraic arguments that seem sound. However, I would like to better understand the technical contributions. Could you clarify what the main novelties and technical challenges are in your proofs? What are the key insights or non-trivial steps that distinguish this work from others?

The clarity and organisation of the proofs could be improved. For instance, assumption 6.3 should be stated much earlier on. The various “Claim”s and proofs of the “Claim”s are non-standard and could be omitted.

I’m not sure if the literature review in this paper is sufficient. For instance, what is the closest prior work to the current paper? I wasn’t able to backtrack.

Can your algorithm be straightforwardly extended to the rank>1 case? I guess no, right?

---

> ### Author Response · Authors · 2025-07-30
> **Response to Reviewer WGnL (1)**
>
> Comment: "The paper reads like ... “Theorem”, “Definition”, “Fact”, “Claim” are overused."
>
> Response: We are sorry to hear that you felt the work was fragmented and difficult to read.
> We hope that the modifications in the revision elevate the paper's readability and organization to your standard.
> With respect to the environment usage, our intent was to appropriately section the work to facilitate the reader's ability in finding results and proofs therein, as well as to encapsulate each of the intermediate proof steps.
> But, we acknowledge that this may not be the visual preference of most readers, so we have removed the "Claim" and "Remark" environments to reduce visual clutter.
>
> ==========================
>
> Comment: "By “nearly-optimal” do you mean ...  support the near-optimality claim."
>
> Response: That's correct. Therefore, when claims of near-optimality are made in the body of the text, we have clarified in terms of the parameter $d$.
>
> ==========================
>
> Comment: "Introduction: “several applications” can you elaborate?"
>
> Response: While the applications of tensor completion is not the topic of our work, in our revision we've now named some specific examples to accommodate.
> We also emphasize the excellent references provided by Kolda \& Bader (2009) and Song et al. (2019) for more information (These references were also in our original submission).
>
> ==========================
>
> Comment: "I understand that ... Better define notation here."
>
> Response: Agreed. In our revision, we have avoided this notation until its definition in the notation section.
>
> ==========================
>
> Comment: "Theorem 1.1 seems unclear ... informal here."
>
> Response: Our decision to place those results together was based on our preference for a visible and concise summarization, but we now understand the confusion that might arise.
> We agree with your suggestion, and have thus replaced that statement with a summary outside of the theorem environment.
>
> ==========================
>
> Comment: "What do you mean by “solves rank-1 completion”? ... better to call the problem “rank-1 tensor completion” throughout."
>
> Response: Your understanding is correct, and we have made a particular effort towards defining the problem of interest ("Rank-1 Completion") in our original submission, and maintaining reference to that problem throughout.
>
> As to your concern about the failure probability, we believe you are speaking in reference to the lower bound.
> In that case, we reiterate that the defined problem requires the tensor to be recovered errorless with probability $\geq 2/3$.
> However, the lower bound shows that (under insufficient sampling) with probability $> 1/3$ the recovered tensor must have error, so that the defined problem cannot be solved in that regime.
> So, to clarify, it's not so much that the failure probability is being used as a measure of recovery quality, per say, but rather it characterizes the regime when the problem is information-theoretically tractable.
>
> To your concern about $m$, we agree that we should reiterate its definition, and in the revision we've included $m$'s definition in the statement of the problem.
>
> We also thank you for your suggestion to calling the problem "Rank-1 Tensor Completion" instead of "Rank-1 Completion" throughout, and have made that change in the revision.
>
> ==========================
>
> Comment: "below Theorem 1.1: incoherence  needs to be defined ... added."
>
> Response: Agreed. We've reinstated a definition from a previous version of this work.
>
> ==========================
>
> Comment: "Definition 1.2: what does CPD [rank of a tensor] stand for?", "Corollary 1.7: can you define what a Hadamard matrix of order d is?"
>
> Response: We apologize for not defining these terms, although we originally felt these terms were sufficiently standard.
> To avoid confusion, we've removed the abbreviation for CPD ("Canonical Polyadic Decomposition"), and included a short footnote to remind the reader of Hadamard matrices.
>
> ==========================

---

> > ### Author Response · Authors · 2025-07-30
> > **Response to Reviewer WGnL (2)**
> >
> > Comment: "Footnote 1-2: what is [complexity class] BPP? why 2/3 rather than for an arbitrarily small constant ?"
> >
> > Response: The complexity class BPP is a canonical class of problems in computational complexity theory. Loosely speaking, it refers to the set of decision problems solved errorlessly with probability at least $2/3$.
> > Its reference is only used to give context for the convention of $2/3$.
> > Note that given an algorithm to solve any of the problems in this class, one can obtain an improved algorithm with smaller error probability and using only a few more samples, namely by the standard method of "repetition-and-majority," as mentioned in the footnote of our original submission.
> > In the revision, we've included a streamlined explanation of the following argument.
> >
> > Suppose you have a randomized algorithm $A$ that returns a correct output with probability $2/3$ on any input $I$, and consumes $m$ samples.
> > Then, one can say $A'$ is the algorithm that, on input $I$, runs $A$ $O(\log(1/\delta))$ times on $I$ and takes the majority.
> > Then $A'$ solves the same problem as $A$ with probability $\geq 1-\delta$, and consumes $O(m \log (1/\delta))$ samples instead of $m$ (hence, a $O(\log (1/\delta))$ overhead is incurred, as mentioned in the original submission).
> > We emphasize that this argument works for any constant strictly larger than $1/2$.
> > Indeed, the usage of $2/3$ is entirely due to convention rather than technical reason, just as in the complexity theory literature.
> >
> > We find this argument is very common in the study of sample complexity (for many problems, not limited to matrix/tensor completion), so we had chosen not to include it in our first submission.
> >
> > ==========================
> >
> > Comment: "Footnotes are overused in the paper ... Footnote 8 should also be defined inline."
> >
> > Response: We thank you for your suggestion, and have both streamlined them and incorporated them into the main body.
> >
> > ==========================
> >
> > Comment: "This paper uses extensive notation, so the Notation paragraph on p.3 should be moved much earlier to help readers."
> >
> > Response: We thank you for your valuable suggestion.
> > In our revision, we have moved up the "Notation" paragraph earlier in the work, also avoiding all nonstandard notation prior.
> >
> > ==========================
> >
> > Comment: "Theorem 1.3: Why introduce q here rather than directly referencing the uniform sampling probability? Also, again, how is the precision of tensor completion defined in this theorem?"
> >
> > Response: To your first question, we'd like to clarify that $q$ is the number of queried entries of the input tensor.
> > In two extremes, we could either define an algorithm which solves the rank-1 tensor completion problem to be that the algorithm outputs the entire tensor correctly, or just a particular entry correctly.
> > These two problems may have completely different time complexities, so we feel it is worthwhile to parameterize the result in terms of $q$.
> > The described algorithm recovers the component vectors as an intermediate step, and computes any particular entry as a product of the associated entries.
> > Therefore, the algorithm runs longer if more entries are of interest, i.e. $q$ is large
> > We hope this clarifies the role of $q$, and how it is separate from the notion of uniform sampling.
> >
> > As to your second question, we re-iterate that the precision of tensor completion in this theorem is exact, i.e. without error, just as we attempted to emphasize in the introduction.
> >
> > ==========================
> >
> > Comment: "Above Definition 1.4: “We believe this is an artifact of our upper bound”. Is there any reason/ related references to back this claim? If not, it is possible the derived lower bound is loose."
> >
> > Response: We mentioned in the "Open Questions" that a slightly more restricted problem (the input tensor is not signed), admits a $O(d\log d)$ sample complexity upper bound, matching the lower bound.
> > The proof is a straightforward corollary of a classic result, but we agree that this should be more visible, particularly in the section you have referenced.
> > This improvement is included in the revision.
> >
> > ==========================
> >
> > Comment: "Definition 1.4: The joint entropy here does not make sense. Shouldn’t it be the conditional entropy $H(A \mid U,S,B)$ = 0? meaning Given knowledge of U, S, and B, there is no uncertainty about A's output?"
> >
> > Response: Your understanding is the intended one, but we are unable to find any discrepancy between the statement you have written and the one in the work (other than using "$;$" instead of "$\mid$", and $\mathcal{A}$ instead of $\mathcal{A}(U,S,B)$, notationally).
> > Does it clarify to note that $\mathcal{A}$ is not a random variable but rather a function, and that the random variable of focus is $\mathcal{A}(U,S,B)$?
> > In any case, we have adopted your notational preference.

---

> > > ### Author Response · Authors · 2025-07-30
> > > **Response to Reviewer WGnL (3)**
> > >
> > > ==========================
> > >
> > > Comment: "Theorem 1.5: I do not understand this lower bound here. What happens with probability 2/3? Since this is a lower bound ie hardness result, the reader would expect that it holds with high probability"
> > >
> > > Response: In terms of the defined problem, since Theorem 1.5 says that any recovered tensor will necessarily have errors with probability $> 1/3$, this means that the problem is not solvable as it requires errorless recovery with probability $\geq 2/3$.
> > > In other words, the recovered tensor will match the original with probability $< 2/3$.
> > > We hope this clarifies why Theorem 1.5 is a hardness result.
> > > We have since added a clarifying statement at the end of Theorem 1.5.
> > >
> > > ==========================
> > >
> > > Comment: "Also in text following this theorem, Assumption 6.3 hasnt been stated at this point, which hinders the readability of the results."
> > >
> > > Response: We originally felt that the statement prior to the referenced point in the text provided a more intuitive and clear idea of the assumption.
> > > We chose to leave its discussion to the pertinent proof section as we felt the full exposition would be too technical and distracting.
> > > However, we agree that the readability could be improved, so we have augmented this high-level explanation.
> > >
> > > ==========================
> > >
> > > Comment: "While I have not verified the proofs in detail, they appear to rely on elementary probabilistic and linear algebraic arguments that seem sound. However, I would like to better understand the technical contributions. Could you clarify what the main novelties and technical challenges are in your proofs? What are the key insights or non-trivial steps that distinguish this work from others?"
> > >
> > > Response: We are pleased that you resonate with our introduction's sentiment about the paper's streamlined analysis.
> > > In the revision of our work, we have emphasized the main novelties and technical challenges at key points, such as during the discussion of: our main results summary, the prior work, the main proof bottleneck, and the conclusion.
> > > In particular, we feel the conclusion is an especially modular and thorough clarification to your request.
> > >
> > > ==========================
> > >
> > > Comment: "I’m not sure if the literature review in this paper is sufficient. For instance, what is the closest prior work to the current paper? I wasn’t able to backtrack."
> > >
> > > Response: There are two main aspects of our work, one being the study of sample complexity, and the other being the focus on rank-1 tensors.
> > > The presence of both these topics are a novel pairing, so the "Prior Work" traces the literature chronologically and separately from each angle.
> > > In the revision, we attempted to emphasize this structure in our literature review.
> > > Moreover, to answer your question, in the final paragraph of the "Prior Work", we note the most related works, Singh et al. (2020) and Stephan \& Zhu (2024), as well as highlight the differences between those papers and ours.
> > >
> > > We'd like to reiterate out that all the results in the first paragraph of the Prior Work section detail the known sample complexity of our studied problem, and that the results in the following two paragraphs detail the study of rank-1 tensors.
> > >
> > > ==========================
> > >
> > > Comment: "Can your algorithm be straightforwardly extended to the rank $>$ 1 case? I guess no, right?"
> > >
> > > Response: Correct. We have incorporated this clarification into the revision.

---

> ### Comment · Reviewer_WGnL · 2025-08-12
>
> I thank the authors for their detailed response, which have addressed most of my questions. The revised manuscript shows improved clarity and flow.

---

> ### Comment · Reviewer_WGnL · 2025-08-12
>
> Thank you for the point-to-point response. I apologise for being unfamiliar with computer science terminology such as BPP.
>
> Your clarifications regarding probability > 2/3 and the number of queried entries $q$ of the input tensor make sense to me.
>
> I think using ';' for conditional notation in entropy is quite uncommon, as ';' typically denotes 'joint' in mutual information.

---

> ### Comment · Reviewer_WGnL · 2025-08-12
>
> I thank the authors for this response, which have addressed my questions. In light of all revisions, I have updated my rating recommending acceptance of this work.

---

### Author Response · Authors · 2025-07-30
**Notice to Revision**

We wish to thank the reviewers and the editor for their valuable criticism, suggestions, and expediency. Below you will find responses to each point raised by the reviewers. Nearly all of these changes resulted in accommodations in the posted revision.

Beyond minor fixes (e.g. typos), this affected only expositional text, with new or heavily-modified portions marked in $\textcolor{blue}{blue}$. For convenience, we summarize the main changes made to the manuscript above, in "Changes Since Last Submission."

---

### Decision · Action_Editor_KJDT · 2025-09-03

**Recommendation:** Accept as is

**Additional Comments:**

The reviewers responded positively to the submission overall. Although some concerns were initially raised, the authors have addressed them adequately through revisions. My own assessment confirms that the key issues have been resolved. Accordingly, we are proceeding with acceptance. That said, please ensure that all requested changes (e.g., clarifications or notational consistencies) are carefully verified and incorporated in the camera-ready version.

**Audience:**

Yes

**Audience Explanation:**

Yes, the paper would likely be of interest to certain members of the TMLR audience. Tensor completion is a commonly used modeling technique in various machine learning applications, and the discussion presented in the paper is therefore relevant.

**Claims And Evidence:**

Yes

**Claims Explanation:**

The paper focuses on a specific characterization of the rank-1 tensor completion problem. This characterization implies a reduced sample complexity. The authors present adequate theoretical results to support their claims.